# Causal phase-dependent control of non-spatial attention in human prefrontal cortex

Jeroen Brus [1,2,6] ✉, Joseph A. Heng [1,2,6], Valeriia Beliaeva [1,2], Fabian Gonzalez Pinto[1,2], Antonino Mario Cassarà [3], Esra Neufeld [3], Marcus Grueschow [4], Lukas Imbach[5] & Rafael Polanía [1,2] ✉

Non-spatial attention is a fundamental cognitive mechanism that allows organisms to orient the focus of conscious awareness towards sensory information that is relevant to a behavioural goal while shifting it away from irrelevant stimuli. It has been suggested that attention is regulated by the ongoing phase of slow excitability fluctuations of neural activity in the prefrontal cortex, a hypothesis that has been challenged with no consensus. Here we developed a behavioural and non-invasive stimulation paradigm aiming at modulating slow excitability fluctuations of the inferior frontal junction. Using this approach, we show that non-spatial attention can be selectively modulated as a function of the ongoing phase of exogenously modulated excitability states of this brain structure. These results demonstrate that non-spatial attention relies on ongoing prefrontal excitability states, which are probably regulated by slow oscillatory dynamics, that orchestrate goal-oriented behaviour.

Is that a predator behind the bush? Is it moving to the left or to the right? When some of these spatially overlapping sensory features are more relevant to guiding behaviour than others, activity in sensory areas representing properties of the attended features is enhanced[1]. This cognitive process is known as non-spatial attention, allowing organisms to orient the focus of conscious awareness towards sensory information that is relevant to a behavioural goal while shifting it away from irrelevant stimuli. Non-spatial attention is commonly subdivided into feature-based attention, focusing on one single feature (such as a direction of motion, colour or orientation), and object-based attention, in which a participant attends to a combination of features (such as an object or a scene). There is consensus that this process is not an intrinsic property of sensory areas but relies on long-range functional interactions with prefrontal structures. A large body of work implicates the inferior frontal junction (IFJ) as a key source of control signals for

both forms of top-down non-spatial attention[2–4]; in contrast, spatial attention has been shown to be governed by a dorsal attention system involving the frontal eye fields and posterior parietal cortex[4,5]. However, the causal mechanisms of top-down regulation of non-spatial attentional control remain unclear.

On the basis of behavioural observations that attentional performance fluctuates over time, rhythmic control has been proposed as a candidate mechanism of attentional regulation[6–8]. Supporting this notion, a study showed that the temporal dynamics of attentional behaviour closely resemble the spectral features of ongoing oscillatory brain activity in prefrontal structures[9]. It was therefore hypothesized that relatively slow and periodic neuronal excitability fluctuations might shape attention and overt behaviour. However, the conclusions from many of these studies have been called into question by suggesting that previously reported rhythmic variations of attentional

[1]Decision Neuroscience Lab, Department of Health Sciences and Technology, ETH Zurich, Zurich, Switzerland. [2]Neuroscience Center Zurich, Zurich, Switzerland. [3]Foundation for Research on Information Technologies in Society (IT'IS), Zurich, Switzerland. [4]Zurich Center for Neuroeconomics, Department of Economics, University of Zurich, Zurich, Switzerland. [5]Swiss Epilepsy Center (Klinik Lengg), Zurich, Switzerland. [6]These authors contributed equally: Jeroen Brus, Joseph A. Heng. ✉e-mail: jeroen.brus@hest.ethz.ch; rafael.polania@hest.ethz.ch

behaviour might be artefacts of the analysis approaches[10]. Moreover, whether ongoing excitability states within prefrontal structures are causally involved in regulating non-spatial attention remains unknown. The role of rhythmic control with a focus on non-spatial attention (that is, without spatial confounds and across both feature-based and object-based attention) also has not been previously established.

Here we attempt to reconcile some of the above-mentioned concerns using a behavioural paradigm coupled with a non-invasive brain stimulation protocol aiming at modulating, with high temporal precision, excitability fluctuations in the IFJ during non-spatial attention in the intact human brain. We emphasize that in our work we do not study the role of endogenous oscillatory fluctuations, but instead study the causal involvement of ongoing excitability states probably driven by slow rhythmic fluctuations (which in our case are exogenously controlled) in top-down attention. It is important to highlight that the causal involvement of prefrontal structures during certain aspects of non-spatial attention has been demonstrated in previous landmark studies using transcranial magnetic stimulation[3]. However, transcranial magnetic stimulation induces only transient disruptions of neural functioning, leaving the role of top-down control through slow fluctuations of the excitability state in prefrontal structures unresolved.

## Spatial and dynamic characterization of non-spatial attention

We designed a behavioural paradigm with the primary goal of inducing a tagged oscillation in the IFJ during non-spatial attention, which would allow us to implement a closed-loop-like simulation protocol to modulate ongoing IFJ excitability states. Participants viewed two spatially overlapping sensory stimuli: (1) a cloud of dots of which a proportion was moving coherently to the left or right side of the screen and (2) images of indoor or outdoor scenes. A series of stimuli went in and out of 'phase coherence' in a sinusoidal manner (at 1.43 Hz) so that they were modulated in visibility over time while changes in luminance and spectral power remained constant (Fig. 1a and Methods). In each trial, the participants were cued to attend one of the two sensory features. At the end of each stimuli stream, the participants were asked to indicate whether the last observed cloud of dots was mainly moving to the left or right (motion cue), or whether the last observed scene was indoor or outdoor (scene cue). The level of sensory evidence in the last stimulus was randomly chosen from one of four predefined levels, allowing us to modulate task difficulty trial by trial, where the smaller the sensory evidence, the more difficult the trial (Methods). We first used both functional MRI (fMRI) (Experiment 1) and high-density electroencephalography (EEG) (Experiment 2) to investigate and validate both the spatial and dynamic involvement of the IFJ in our non-spatial attention task. Crucially, we implemented a control 'no-attention' task that contained identical visual input as the non-spatial attention task, but where the stream of fluctuating sensory information was behaviourally irrelevant (Supplementary Fig. 1 and Methods).

In Experiment 1, we found that the bilateral IFJ was the most active prefrontal brain area (in terms of both cluster size and peak $Z$ score) in the attention task compared with the no-attention task for each sensory modality (peak $Z_{motion}$ = 5.9, $Z_{scene}$ = 6.1, $P < 0.001$, $P < 0.05$ cluster corrected; Fig. 1b), with a high degree of overlap across the two sensory modalities (conjunction analysis $Z > 2.6$, $P < 0.05$ cluster corrected; Fig. 1b). Peak activations in the contrast of attention > no attention occurred at Montreal Neurological Institute (MNI) coordinates 54, 10, 36 and −42, 2, 30, which fit well with the location of the IFJ in the literature[11] (Supplementary Fig. 2 and Supplementary Tables 1–5). The contrast of attention to motion versus attention to scene showed that the bilateral middle temporal complex was selectively active during motion-cued trials (peak $Z$ = 5.7, $P < 0.001$, $P < 0.05$ cluster corrected; Fig. 1b), and this result was accompanied by significant psychometric performance for motion evidence (random effects estimate ($\beta_{RFX}$) = 11.0; 95% confidence interval (CI), (9.0, 13.0); Markov chain

Monte Carlo $P$ ($P_{MCMC}$), <0.001; Fig. 1c,d), but not for scene evidence ($\beta_{RFX}$ = 0.1; 95% CI, (−1.9, 2.1); $P_{MCMC}$ = 0.45). In contrast, the parahippocampal place area (PPA) was more active during scene-cued trials (peak $Z$ = 4.6, $P < 0.05$ cluster corrected; Fig. 1b), and this result was accompanied by significant psychometric performance for scene evidence ($\beta_{RFX}$ = 19.0; 95% CI, (17, 21); $P_{MCMC}$ < 0.001; Fig. 1c,d), but not for motion evidence ($\beta_{RFX}$ = 1.2; 95% CI, (−0.8, 3.2); $P_{MCMC}$ = 0.12). As a sanity check, we show the main effects of the task (that is, without contrasting attention versus no-attention states) and found that most of the visual cortex was similarly active when paying attention to motion and scenes (Supplementary Fig. 3), suggesting the specificity of top-down control involving the fronto-parietal network, which prominently engages the IFJ (Fig. 1b).

We next investigated whether the IFJ was indeed rhythmically tagged to the stimulus visibility, and if so, whether this was more prominent during attention or during no attention. We computed the debiased weighted phase lag index (dWPLI) at 1.43 Hz between the sensor data and the visual stimulation signal. This measure captures how much the EEG is tagged to the visual stimulation. First, we compared these values between the attention and no-attention tasks across electrodes and found clusters where the dWPLI was higher in the attention task ($T_{max}$ = 4.81, $P_{cluster}$ = 0.001 and $T_{max}$ = 4.51, $P_{cluster}$ = 0.001 for the occipital and frontal clusters, respectively; $P < 0.01$ whole-brain cluster corrected; Fig. 1e) (for the time series of an example participant, see Fig. 1f; for the dWPLI across frequencies, see Supplementary Fig. 4). We then computed the dWPLI at the source level by conducting a whole-brain analysis (Methods). Without contrasting attention versus no-attention states, we found that posterior brain areas get entrained to the frequency of the visual input, where the degree of entrainment is higher for visual areas (Fig. 1g). Contrasting attention and no attention, we found a significant cluster located near the left IFJ ($T_{max}$ = 4.14, $P_{cluster}$ = 0.040; Fig. 1h,i). Despite the well-known lack of spatial precision resulting from EEG source analyses, we found a remarkable degree of overlap between the resulting significant EEG and fMRI clusters (Supplementary Fig. 5 and Supplementary Table 6). The lateralized prefrontal cluster is located in the vicinity of the IFJ, and given the low spatial resolution of EEG, it is likely that this cluster is related to the IFJ, which is clearly activated following our fMRI analyses. To estimate the latency of sensory responses in the IFJ during the attention task, we extracted the relative phase lag between the frequency-tagged response and the stimulus on the screen. The average phase lag of the IFJ was 157 ms, which was shifted by 60 ms relative to early sensory areas (Fig. 1j); this is probably related to synaptic delays between areas and roughly follows previous reports[2]. At the behavioural level, these results were accompanied by a significant impact of motion evidence on performance when motion was cued ($\beta_{RFX}$ = 6.3; 95% CI, (4.3, 8.3); $P_{MCMC}$ < 0.001; Fig. 1k,l), but not of scene evidence ($\beta_{RFX}$ = −1.4; 95% CI, (−3.4, 0.6); $P_{MCMC}$ = 0.08). Conversely, when scene was cued, psychometric performance was significant for scene evidence ($\beta_{RFX}$ = 19.1; 95% CI, (17.1, 21.1); $P_{MCMC}$ < 0.001; Fig. 1k,l), but not for motion evidence ($\beta_{RFX}$ = 1.3; 95% CI, (−0.7, 3.3); $P_{MCMC}$ = 0.09).

Additionally, we investigated whether some of the above-mentioned differences in top-down attentional control by the IFJ could be related to stronger oculomotor engagement in our task. Analyses of eye tracking data show that there is no significant difference between the number of saccades or microsaccades in the motion, scene or no-attention condition; therefore, differences in eye movements cannot explain the differences in brain activity (Supplementary Fig. 6). Taken together, our set of behavioural and imaging analyses strongly suggest the involvement of the IFJ during non-spatial attention in our task and the selectivity of sensory areas for each relevant feature.

## Exogenous control of IFJ top-down attention

Having established rhythmic IFJ engagement during non-spatial attention in our task, the fundamental question we asked is whether the

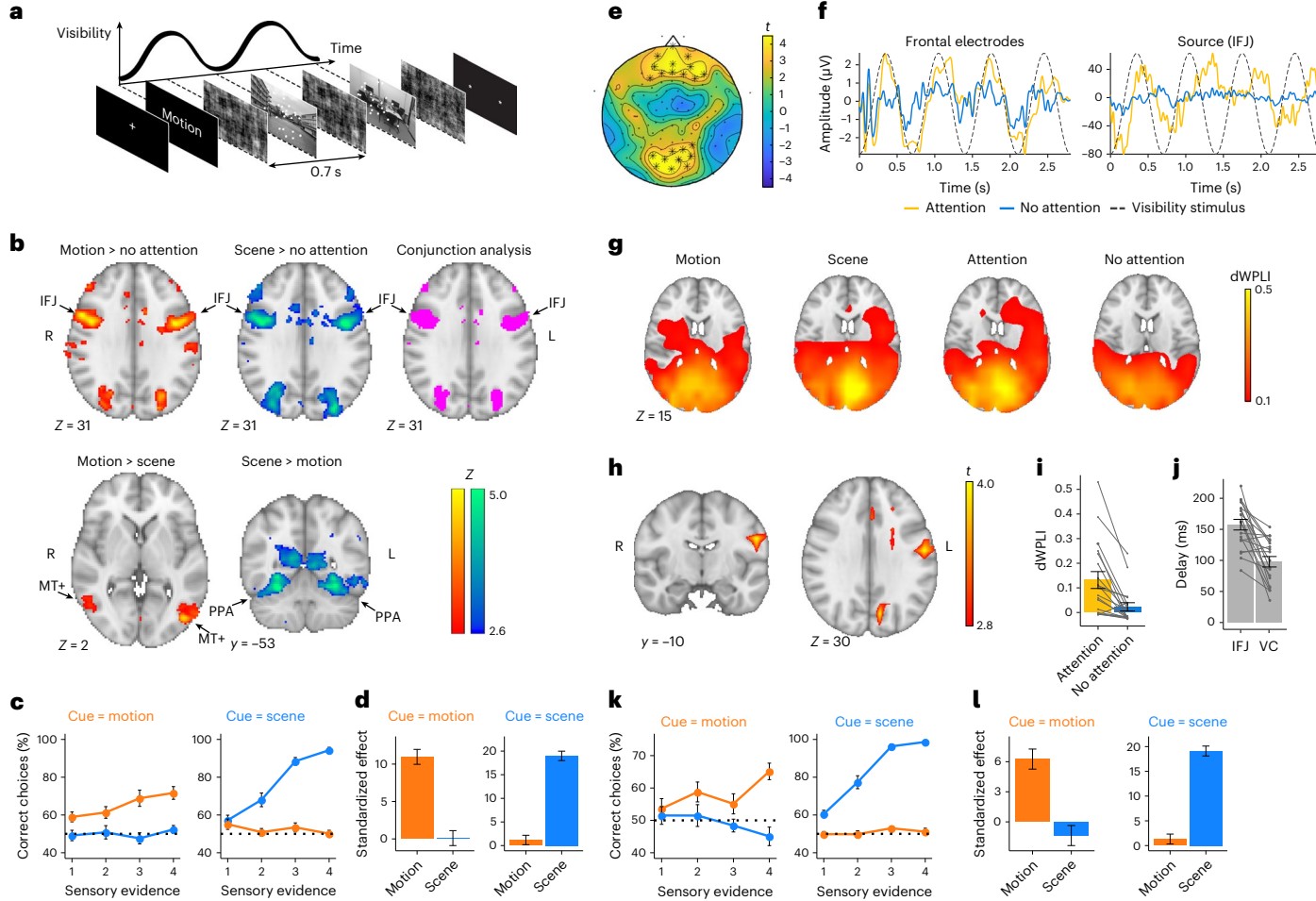

**Fig. 1 | fMRI and EEG paradigm, Experiments 1 and 2. a**, Example display of one trial. After the attentional cue, a sequence of four to seven compound stimuli was presented following a sinusoidal rhythm through time at 1.43 Hz. The participants responded with a button press, taking only the last motion/ scene stimulus into account. If motion was cued, the participants pressed left for leftward motion and right for rightward motion; if scenes were cued, they pressed left for indoor and right for outdoor scenes. **b**, fMRI results ($n = 20$), Experiment 1. Attention to motion and scenes versus no attention shows that the IFJ activates bilaterally. A contrast of attention to motion versus scenes shows that the area associated with motion perception, the middle temporal complex (MT+), activates. The inverse contrast shows that the area sensitive to scene recognition, the PPA, activates. The images were thresholded at $Z > 2.6$ and whole-brain cluster corrected at $P < 0.05$. **c**, Behavioural results, Experiment 1. The participants used the motion evidence when cued to pay attention to motion (orange) and the scene evidence when cued for scene (blue) and crucially ignored the irrelevant sensory feature. The error bars denote ±s.e.m. **d**, Standardized coefficients of a multifactor logistic regression of task performance as a function of evidence levels show that the participants were significantly influenced by the cued evidence (cue = motion $\beta_{RFX} = 11.0$; 95% CI, (9, 13); $P_{MCMC} < 0.001$; cue = scene $\beta_{RFX} = 19.0$; 95% CI, (17, 21); $P_{MCMC} < 0.001$) and not distracted by the irrelevant sensory feature. The standardized effect represents the expected value of the corresponding posterior $\beta$ estimate ± s.d., divided by its standard deviation. **e**, EEG results ($n = 19$), Experiment 2. We computed the dWPLI at 1.43 Hz between

the sensor data and the visual stimulation signal. The topoplot shows the statistical difference in the dWPLI between the attention and the no-attention tasks at the sensor level, indicating that frontal and occipital electrodes are more entrained to the visual stimulus during the attention task. Starred electrodes represent significant electrodes (cluster corrected at $P < 0.01$; Methods). **f**, Event-related potential for the first four periods of the visual stimulus of an example participant at the scalp sensor (left) and source (right) levels. Sensor-level signals are shown for the frontal cluster of electrodes with a higher dWPLI during the attention versus no-attention task (see **e**). **g**, The dWPLI between the EEG data during the four first periods of the visual stimulus and the 1.43 Hz visual signal was computed. The dWPLI values show that a wide area of the visual cortex gets tagged to the frequency of the visual stimulation. **h**, The dWPLI between the beamformed signals of each voxel and the visual stimulus was computed for attention versus no attention. The maps show the statistical difference between the two attention conditions, revealing the left IFJ to be tagged to the degree of stimulus visibility during attention trials (whole-brain cluster corrected at $P < 0.01$; Methods). **i**, dWPLI values of the IFJ cluster in **g**. The data are presented as mean values ± s.e.m. **j**, The prefrontal cortex and visual cortex (VC) activate around 157 and 98 ms after the visual stimulation, respectively. The error bars denote ±s.e.m. **k,l**, Task performance and standardized coefficients of a multifactor logistic regression of task performance in Experiment 2 replicate the effects observed in Experiment 1 (see **c,d**). The error bars denote ±s.e.m.

slow fluctuations of the excitability state exogenously induced in the IFJ are causally related to top-down control. A key feature of our behavioural paradigm is that it allows us to predict latencies at which neural excitability for sensory processing is high. We hypothesized that boosting periods of predicted high-excitability states in the IFJ would promote perceptual discriminability performance for the cued sensory feature. Conversely, downregulating periods of predicted

high-excitability states would hinder behavioural performance (Fig. 2b). To test this hypothesis, we employed transcranial alternating current stimulation (tACS), a technique that has the potential to establish a causal link between oscillatory patterns—modulated or induced[12–17]—at the targeted brain structure and the resulting behaviour. We applied tACS targeting the IFJ bilaterally using a ring electrode configuration to increase the focality of the induced electric

fields (Fig. 2a, Supplementary Fig. 7 and Methods). We applied 5% EMLA cream under the stimulation electrodes, allowing us to reduce somatosensory effects, increase stimulation intensities (up to 4 mA peak-to-peak; Methods) and thereby increase the chances of oscillatory neuromodulation. We applied tACS at the same sensory tagging frequency (1.43 Hz), but crucially, the presentation of sensory stimuli was precisely synchronized to the tACS waveform in one of two ways in each trial. First, the peak of anodal stimulation of the centre electrode (defined as the peak of the waveform) coincided with periods of high sensory excitability (the 'in-phase' condition, while considering the delays estimated in the EEG experiment; Methods), which we expected to result in attentional improvements because anodal stimulation is thought to increase the excitability states of the targeted cortical structure[18]. Second, the peak of cathodal stimulation of the centre electrode (which we define as the trough of the waveform) coincided with periods of high sensory excitability (the 'out-of-phase' condition), which should result in attentional hindering by reducing the cortical excitability states of the IFJ[18] (Fig. 2b and Supplementary Fig. 8).

In one of two lab visits, the participants received in-phase tACS for one of the two sensory cues (attending to motion or scene, Experiment 3a) and received out-of-phase tACS for the other sensory cue. The stimulation conditions were switched for each sensory cue in the second lab visit (Experiment 3b, Methods). We first investigated whether, during the stimulation-on trials, in-phase stimulation improved behavioural performance relative to out-of-phase stimulation. In line with our hypothesis, we found that, compared with out-of-phase stimulation, in-phase stimulation improved sensory discrimination performance when motion was cued (interaction of sensory evidence × stimulation condition, $\beta_{RFX} = 2.8$; 95% CI, (0.8, 4.8); $P_{MCMC} = 0.004$; Fig. 2e); however, we did not find a significant effect when scenes were cued ($\beta_{RFX} = 1.3$; 95% CI, (−0.7, 3.3); $P_{MCMC} = 0.086$). Post hoc analyses revealed that discrimination performance improved in the hypothesized direction for motion discrimination at the highest levels of difficulty (one-tailed paired-samples Wilcoxon test; $V = 417$, $P < 0.001$, $r = 0.55$, 95% CI, (6.2, ∞) and $V = 420$, $P = 0.042$, $r = 0.28$, 95% CI, (0, ∞) for levels 1 and 2, respectively; Fig. 2d) and for scene discrimination at the highest level of evidence ($V = 328.5$; $P = 0.023$; $r = 0.35$; 95% CI, (0, ∞)). We employed the same multifactor regression to investigate whether stimulation exerted influences on the distractor (non-cued) sensory feature. We found no effect of stimulation in either task ($P_{MCMC} > 0.16$ in both tasks; Fig. 2e). This indicates that modulations of ongoing IFJ fluctuations induced by our stimulation protocol exclusively affect attention to the relevant (cued) sensory feature.

## Dynamic evolution of IFJ top-down control modulations

The previous analyses were carried out during stimulation-on periods but do not allow interpreting whether these effects emerge exclusively during online stimulation, how they temporally evolve and how these compare to periods without stimulation. To investigate this, we analysed the temporal evolution of the in-phase versus out-of-phase stimulation effects (initially across sessions; Methods). When motion was cued, we found that the stimulation-induced attentional modulations emerged exclusively during the stimulation-on periods and vanished immediately after the stimulation was switched off ($P < 0.05$ cluster corrected; Fig. 2g), and were in the correct direction but not significant when attention to scenes was cued. For a comparison of naturally occurring fluctuations in performance, see Supplementary Fig. 9. While these analyses reveal the robustness of the effects (when motion is cued and despite potential behavioural variability across sessions), these results do not allow us to conclude whether the stimulation-induced across-session modulations are driven by in-phase stimulation, out-of-phase stimulation or both. To investigate this, we analysed the evolution of the stimulation effects within a single stimulation session relative to baseline periods of no stimulation (Methods). We found that out-of-phase stimulation robustly hindered discrimination performance exclusively during stimulation-on periods when motion was cued ($P < 0.05$ cluster corrected; Fig. 2h), but this effect was not significant during in-phase stimulation ($P > 0.05$ cluster corrected; Fig. 2h), and once again, these effects vanished immediately after the stimulation was switched off. Crucially, the interaction of motion evidence × stimulation condition was robustly significant in the hypothesized direction exclusively during the stimulation-on periods ($P < 0.05$ cluster corrected; Fig. 2h). Once again, these effects were not present for the distractor feature (Supplementary Fig. 10). Aligning periods of high-excitability states in the IFJ with electric fields thus modulates non-spatial attentional behaviour, and these effects are robust for motion perception.

## Top-down control specifically affects sensory processing

While our brain stimulation protocol appears to induce robust attentional influences in motion discrimination performance, these results do not clarify whether these behavioural modulations are indeed specific to boosting the perception of sensory evidence. We employed the drift-diffusion model, a well-established mathematical model of human choices that allows the possibility of disentangling how the manipulation of IFJ excitability states affects latent variables corresponding to distinct components of the decision process (Methods).

**Fig. 2 | Temporal alignment of tACS over the IFJ modulates sensory perception (Experiment 3, $n = 37$). a,** Two concentric electrode pairs were placed over the left and right IFJ, reaching relatively focused peak electric fields of ~0.5 V m⁻¹ (Supplementary Fig. 7). **b,** The tACS current followed a sinusoidal function applied either in-phase relative to the visual tagging response or out-of-phase with a phase lag of 180° relative to the visual tagging response. **c,** The percentage of correct trials at different difficulty levels shows that the participants used the cued sensory evidence and ignored the irrelevant stimuli. The data are presented as mean values ± s.e.m. **d,** Participants performed better in the in-phase tACS condition than in the out-of-phase condition when they were cued to pay attention to motion mostly at the hardest difficulty levels (one-tailed paired samples Wilcoxon test; $V = 417$, $P < 0.001$, $r = 0.55$, 95% CI, (6.2, ∞) and $V = 420$, $P = 0.042$, $r = 0.28$, 95% CI, (0, ∞) for levels 1 and 2, respectively) and for scene discrimination at the highest level of evidence ($V = 328.5$; $P = 0.023$; $r = 0.35$; 95% CI, (0, ∞)). The data are presented as mean values ± s.e.m. **e,** A linear mixed-effects model reveals that in the motion trials (besides the main effect of motion evidence; $\beta_{RFX} = 10.6$; 95% CI, (8.6, 12.6); $P_{MCMC} < 0.001$), there is a significant interaction effect between motion evidence and stimulation condition ($\beta_{RFX} = 2.8$; 95% CI, (0.8, 4.8); $P_{MCMC} = 0.004$), with no effect of the irrelevant sensory feature.

In scene trials, only the main effect of scene is significant ($\beta_{RFX} = 13.7$; 95% CI, (11.7, 15.7); $P_{MCMC} = 0.001$). The standardized effect represents the expected value of the corresponding posterior $\beta$ estimate ± s.d., divided by its standard deviation. **f,** Computational modelling analysis based on the drift-diffusion model reveals that tACS-induced behavioural modulations when motion is cued are specifically related to enhancing the rate of sensory evidence ($\beta_{RFX} = 2.6$; 95% CI, (0.6, 4.6); $P_{MCMC} = 0.0018$) while leaving all other parameters unaffected. The standardized effect represents the expected value of the corresponding posterior $\beta$ estimate ± s.d., divided by its standard deviation. **g,** A moving-window analysis shows that the effect of the stimulation is online. The grey shaded area indicates the windows for which stimulation was turned on. The lines indicate the expected values, and the shaded areas around the lines indicate ±1 s.d. of the posterior estimate of the interaction evidence × stimulation. The black bar at the top indicates $P < 0.05$ cluster-corrected effects. **h,** We found that for motion-cued trials (left), out-of-phase stimulation significantly hindered performance ($P < 0.05$ cluster corrected). The lines indicate the expected values, and the shaded areas around the lines indicate ±1 s.d. of the posterior estimate of the interaction evidence × stimulation.

If it is true that IFJ excitability modulations specifically affect the degree of efficiency at which sensory areas accumulate sensory evidence, then we would expect opposing stimulation protocols to selectively affect the rate of sensory evidence accumulation. In line with our hypothesis, we found that in-phase stimulation during motion-cued trials exhibited a higher rate of sensory evidence accumulation (interaction of sensory evidence × stimulation condition, $\beta_{RFX} = 2.6$; 95% CI, (0.6, 4.6); $P_{MCMC} = 0.0018$; Fig. 2f), while leaving all other latent variables unaffected ($\beta_{RFX} < 1.3$; $P_{MCMC} > 0.09$).

Crucially, we investigated whether some of the above-mentioned differences in the modulation of top-down attentional control were related to our non-invasive brain stimulation intervention inducing oculomotor modulations. Analyses of eye tracking data show that there is no significant difference between the number of microsaccades in the different brain stimulation conditions (Supplementary Fig. 6). Together, our oculomotor and modelling analyses provide evidence that stimulation-induced attentional modulations are specifically related to boosting the degree of efficiency at which sensory areas accumulate sensory evidence[19,20].

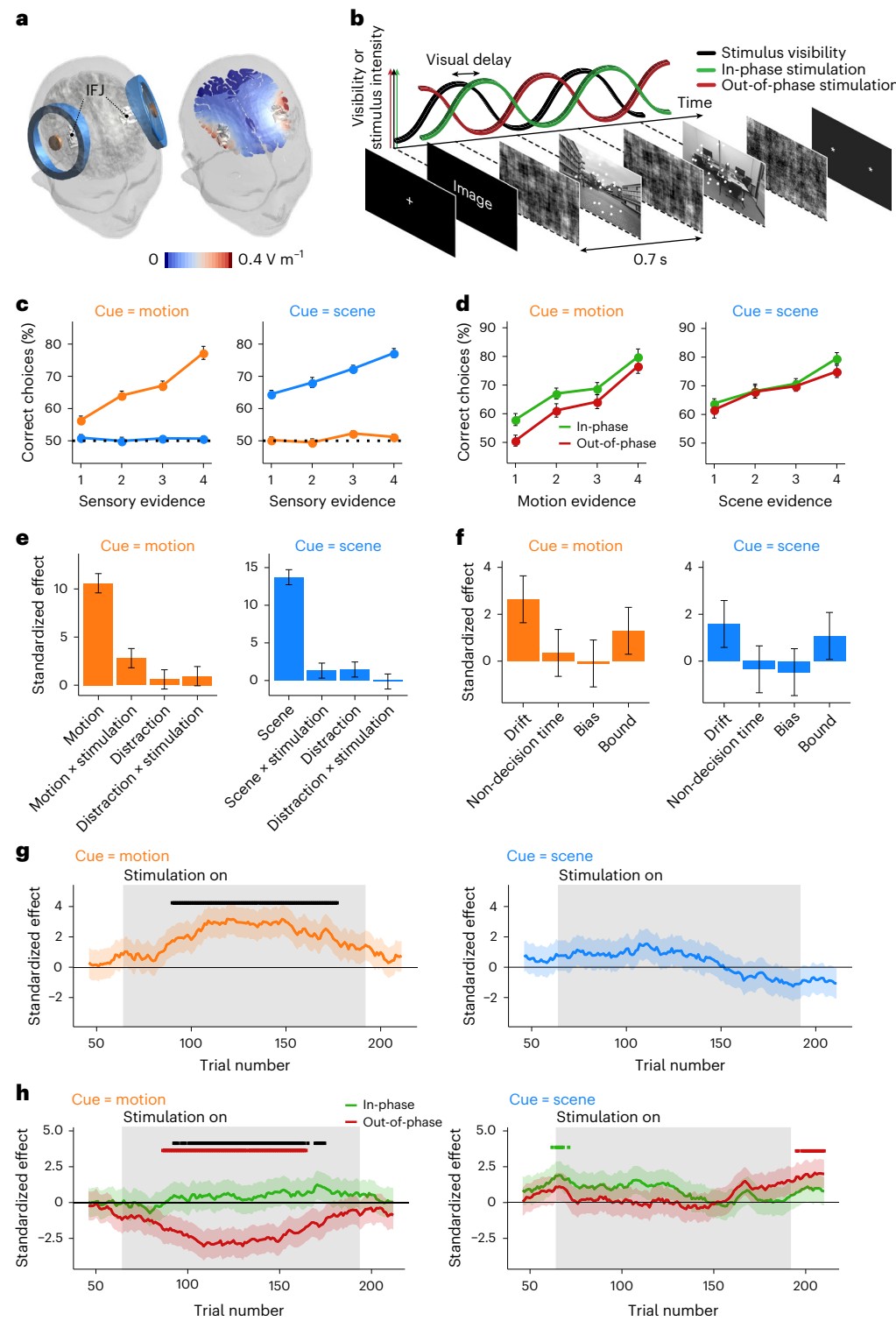

## Non-invasive phase-dependent control of non-spatial attention

The next question we asked is whether the stimulation-induced attentional modulations necessarily require sensory tagging of the IFJ to rhythmic sensory manipulations. We also reasoned that the relatively weak effect for scenes in Experiment 3 might be due to the multidimensional and non-local nature of the scene stimuli. In other words, because there is a larger activation area for scene recognition (Fig. 1b), it might be harder to find the optimal timing of the stimulation if there is some degree of variability in the reaction of the cortical responses to sensory tagging across participants during the presentation of more complex sensory stimuli. To study these issues, we performed a new experiment (Experiment 4), where in each trial we presented a single stimulus that went in and out of phase coherence (Fig. 3a and Methods). An additional feature in Experiment 4 is that we not only stimulated in-phase or out-of-phase (as in Experiment 3) but also applied tACS at six different delays relative to the presentation of the sensory stimulus (Fig. 3a). This allowed us to investigate whether non-spatial attentional modulations would fluctuate as a function of the phase of the tACS-induced electric field. We found that the ongoing phase of the tACS signal induced significant modulations of behavioural performance when motion was cued (standardized estimate of the amplitude modulation effect ($z_A$), 2.1; 95% CI, (0.1, 4.1); $P = 0.016$; permutation tests; Fig. 3c,d and Methods). This effect was smaller in overall effect-size terms but robustly significant when scene was cued ($z_A = 1.8$; 95% CI, (−0.2, 3.8); $P = 0.039$; permutation tests; Fig. 3c,d). We estimated the optimal timing of the peak of the electrical stimulation to be 7° after the peak of visual stimulation for motion trials and 39° for scene trials (136 and 74 ms before the peak activation of the IFJ for motion and scene trials, respectively; Supplementary Fig. 11). Additionally, we performed a Bayes factor analysis to test the statistical evidence of phasically modulated discrimination performance relative to the null (defined here as $BF_{10}$, with the null calibrated on the basis of the null distribution; Methods). We found $BF_{10} > 100$ for motion and $BF_{10} = 41$ for scenes, which indicates 'extreme' and 'very strong' evidence against the null model[21]. The results of this experiment allow us to conclude that, first, continuous rhythmic sensory tagging is not necessary for inducing IFJ excitability modulations; and, second, non-spatial attention is related to excitability states of the IFJ, which can be modulated as a function of exogenously applied electric fields.

## tACS-induced effects are not related to peripheral nerve stimulation

We conducted a new experiment (Experiment 5) to test whether the effects of tACS on non-spatial attention observed in Experiment 4 are (1) specific to the IFJ, (2) not due to our tailored design to induce a generalized oscillatory sensory tagging in the brain, (3) not due to transcutaneous stimulation of peripheral nerves[16,22] and (4) not related to potential marginal influences of the electric field potentially reaching sensory areas. We identified (on the basis of our neuroimaging data experiments) and stimulated a different brain structure from the IFJ that was in principle not related to non-spatial attention. The cortical area that we selected as the control target was the vertex (the Cz location of the 10–20 EEG coordinate system, a structure that is typically used as an active control site in non-invasive brain stimulation investigations studying higher cognitive functions[23]; Fig. 3e). All other experimental parameters were equal to those of Experiment 4.

First, we confirmed that the electric fields in this active control condition do not greatly influence the IFJ, PPA and V5. We found that the electric fields are virtually ineffective in these cortical areas (<0.1 V m⁻¹ for all voxels in the regions of interest; Supplementary Fig. 7). Second, in the tACS behavioural experiment, we found no significant modulations of behavioural performance as a function of the phase of the tACS-induced electric field for motion ($z_A = 0.3$; 95% CI, (−1.7, 2.3); $P = 0.37$; permutation tests) or for scenes ($z_A = 0.6$; 95% CI, (−1.4, 2.6); $P = 0.28$; permutation tests). Moreover, the strength of the evidence favouring the null on the basis of the Bayes factor analyses (defined here as $BF_{01}$) revealed $BF_{01} = 3.5$ for motion and $BF_{01} = 3.3$ for scenes, which indicates 'substantial' evidence for the null model[21]. This active control experiment thus suggests that the modulatory effects of tACS on non-spatial attention observed in Experiment 4 are indeed related to the stimulation of the IFJ and not due to the above-mentioned alternative explanations.

## Discussion

We developed a behavioural paradigm alongside a closed-loop non-invasive brain stimulation protocol that allowed us to predict and modulate with high temporal precision the IFJ excitability states during a non-spatial attention task. When the IFJ was predicted to be in a high-excitability state, modulating it with tACS resulted in non-spatial attentional performance alterations. These effects were robust for motion evidence and replicated in a second experiment in which attentional modulations did not require a steady IFJ sensory tagging.

While in general the effects for scenes were in the hypothesized direction, they were not significant in the experiment with fixed in- and out-of-phase timings as identified for a different population sample in the EEG experiment. However, in Experiment 4 we show significant modulation of attention to scenes. It could be that the variety of stimulation timings in Experiment 4 makes it less sensitive to inter-individual differences. Given that sensory evidence for scenes is not a unidimensional sensory feature and engages various features and a large

**Fig. 3 | Phase-dependent influence of IFJ tACS but not Cz tACS on non-spatial attention (Experiments 4 and 5). a**, In this experiment, we introduced six stimulation delay conditions. The phase delays between the electrical and visual stimulation are evenly spaced over one period of stimulation. We fit a sinusoidal function to the modulation of feature-based attention as a function of phase delay; the amplitude of this function is the parameter of interest. **b**, In Experiment 4 ($n = 37$), the centre of the electrodes was placed over the IFJ. **c**, Since amplitude is a positive metric, we investigated its significance level by randomly shuffling all stimulation delay labels within participants and comparing the resulting distribution of estimated amplitudes with the estimated amplitude of the sinusoidal fit of the original data (vertical dashed line). We found that the amplitude of the fit of the original data is larger than 98.4% of the amplitudes of the generated distribution for motion trials and 96.1% for scene trials. Bayes factor analysis showed 'extreme' and 'very strong' evidence against the null model ($BF_{10,motion} > 100$ and $BF_{10,scene} = 41$) **d**, The $Z$ scores of the empirical amplitudes as compared to the distribution of amplitudes expected to be found by chance is 2.1 for motion trials and 1.8 for scene trials. The error bars indicate ±1 s.d. **e**, In Experiment 5 ($n = 37$), the centre of the electrodes was

placed on the location of the Cz electrode of the 10–20 EEG coordinate system. **f**, The control experiment shows that stimulating the motor cortex leads to no significant modulation of feature-based attention to either motion or scenes (the empirical amplitudes are larger than 59% and 67% of the generated distribution of amplitudes, respectively). The effects of stimulation therefore cannot be attributed to the stimulation of an unrelated cortex or peripheral nerves. We found Bayes factors of $BF_{01,motion} = 3.5$ and $BF_{01,scene} = 3.3$, which indicates 'substantial' evidence for the null model. **g**, The $Z$ scores of the control experiment are 0.3 for motion and 0.6 for scenes. The error bars indicate ±1 s.d. **h**, The sinusoidal function of performance versus stimulation delay in Experiment 4, with the estimated population-level parameters represented as lines; the shaded area indicates ±1 s.d. The dots represent the individual data for each participant per stimulation delay condition after being aligned for individually estimated phase delays and intersects. The vertical green and red bars indicate the time windows of best and worst performance, respectively. **i**, Psychometric curves of the highest-performance phase delay (green) and worst-performance phase delay (red). The data are presented as mean values ± s.e.m. **j,k**, Similar to **h,i**, but for Experiment 5.

portion of the ventral visual stream (unlike motion perception), our stimulation protocol used in Experiment 3 might require more specific sensory features to be more effective. Future experiments aiming at modulating more complex sensory stimuli might profit from an individualized approach—for example, by first performing an EEG experiment and estimating optimal timings of the stimulation per participant. As another option, on the basis of the observation that

most stimulation-induced effects in our study were in the hypothesized direction, it is tempting to speculate that increasing electric fields in the target area may result in more effective neural modulations and consequently more effective behavioural influences[14,24].

The IFJ attention maps in our imaging analyses were obtained on the basis of the contrast of an attention task (focused on either motion or scenes) versus a no-attention task (in which the participants saw

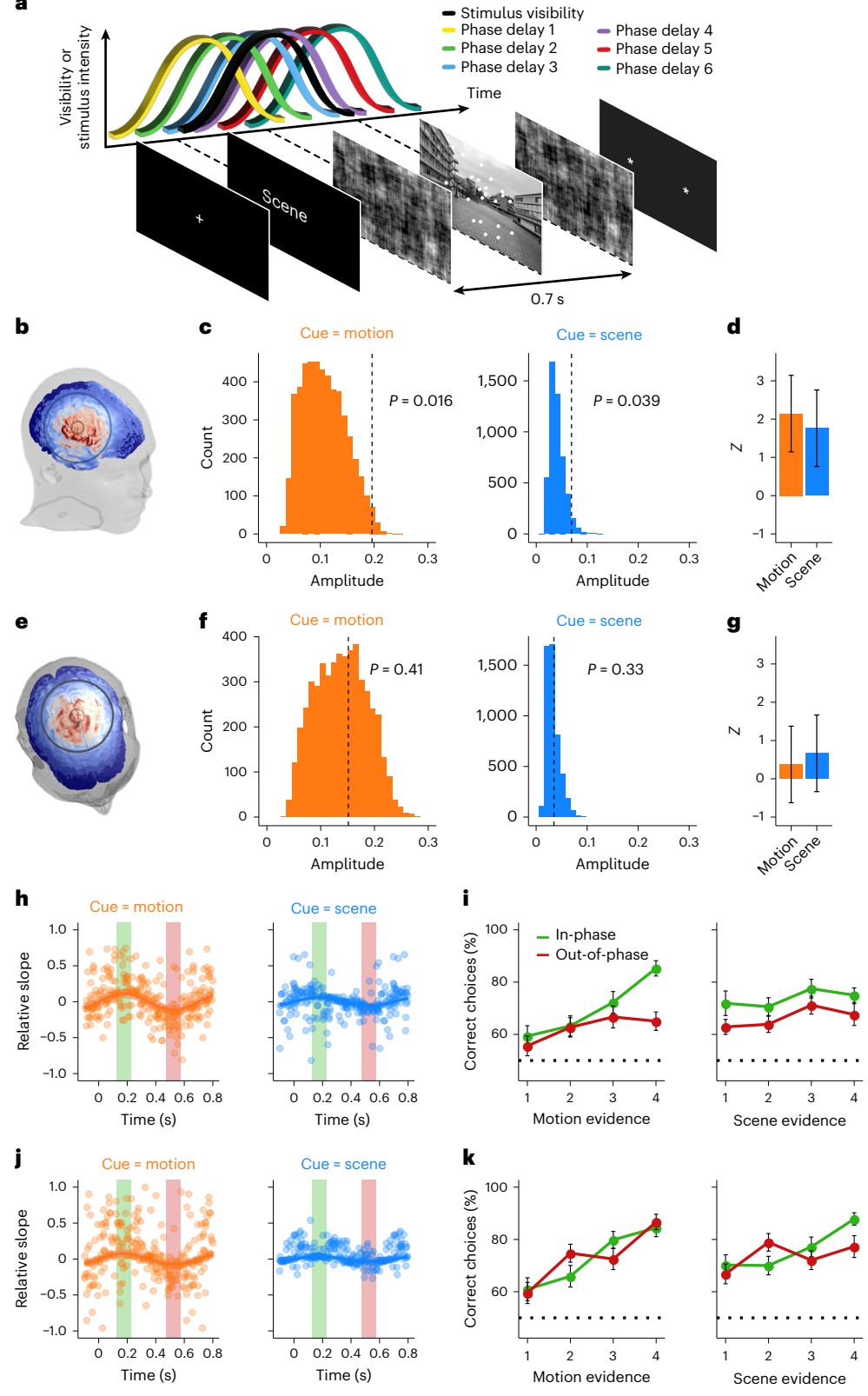

the same visual stimuli, but only reported the incidental rotations of a fixation cross). Given that we performed the no-attention task first, we cannot rule out the possibility that our imaging results can be explained by order effects. However, note that performing the no-attention task first (before explaining the attention task) was necessary to make sure that the participants would not start performing the attention task early as a way of practising and preparing during the no-attention task. We also cannot exclude the possibility that the results of our IFJ causal manipulations are not specific to central attention as opposed to peripheral attention. This is an interesting possibility that should be pursued in future studies—for example, by adapting the current no-attention version of the task by allowing the button press cue to appear at a random location on the screen. Lastly, although we used the relatively focal ring electrodes, and electric field modelling shows that we mainly targeted the IFJ (Supplementary Fig. 7), other tissues in the neighbourhood of the IFJ are also stimulated. We cannot rule out with certainty that we directly or indirectly stimulated other brain areas that could be causally involved in feature-based attention. Future advances in non-invasive neuromodulation selectivity could allow us to determine more precisely which areas are driving the results.

Our results show that the IFJ is causally involved in top-down non-spatial attention, which is in line with what is known about the anatomical and functional connectivity of the IFJ. It has been shown that the IFJ has systematic fMRI co-activations with the ventral visual pathway, areas that are involved in high-level non-spatial attention[11]. On the basis of maps of probabilistic connectivities[25], it has been shown that the IFJ has a high connection probability with both the fusiform face area and the PPA[2]. Interestingly, the coherence between these areas increases in a tagging frequency as well as in high-gamma frequencies when attention to houses (PPA) and attention to faces (fusiform face area) are exerted[2]. Furthermore, studies of spontaneous activity as measured with magnetoencephalography recordings show that the IFJ has a strong power coupling with the ventral visual stream in delta, beta and gamma oscillations[26]. It is likely that we target the coherence of these types of top-down communication channels with our tACS paradigm via the modulation amplitude coupling fluctuations. It will be exciting to see future research pinpointing whether our tACS paradigm exerts its influence through modulation of the coherence of the tagging frequency or other frequencies, such as high-gamma, using imaging methods more sensitive to high-frequency oscillations such as magnetoencephalography. These findings could then be compared to studies of spatial attention, since the IFJ is often contrasted to the frontal eye field, which connects more strongly to the dorsal visual stream and is thought to have a similar attentional control function in spatial attention[4,11,26].

Temporal manipulations of sensory evidence in recent behavioural and neuroimaging studies have led researchers to hypothesize that slow periodic neuronal excitability fluctuations in prefrontal structures shape the temporal dynamics of attention[6–9]. These studies show that in both spatial and non-spatial attention, locations and features are rhythmically sampled. This means that even during instructed (or also perhaps intended) sustained attention, there are periods of enhanced and diminished perceptual sensitivity. One possibility is that this helps higher-order brain areas optimally use their limited neural resources by segregating in time their functional connectivity with either sensory or motor areas. Periods in which functional connectivity with sensory areas is relatively high correspond to sampling windows and high perceptual sensitivity, while periods with high connectivity to motor areas correspond to periods in which attention is switched to a different location, feature or object[27]. However, the hypothesis that periodic excitability fluctuations shape the dynamics of attention was questioned in a recent study suggesting that evidence for attentional rhythmic control is far from definitive due to statistical weaknesses in the analysis approaches[10].

While we did not study the role of endogenous fluctuations but controlled them exogenously, our paradigm and results provide evidence that prefrontal excitability states are causally related to guide top-down attention. We acknowledge that with our paradigm we cannot distinguish between modulating endogenous neural activity and modulation of phasic activity on top of the exogenously controlled oscillation. However, irrespective of this consideration, our results support the theory that non-spatial attention relies on ongoing prefrontal excitability states, which are probably regulated by slow oscillatory dynamics that guide goal-oriented behaviour. Following up on our predator example, our findings indicate that if the direction of the predator's movement behind the bush is the relevant feature, high-excitability states in prefrontal structures regulating top-down attention would promote correct discrimination of the predator's direction of movement.

It is important to highlight that the IFJ might be involved in more processes than just non-spatial attention. It is proposed that many prefrontal regions work together in a general multiple-demand system that is particularly important when solving novel tasks that require, for instance, fluid intelligence[28]. What role the IFJ plays within this multiple-demand system and whether this is also related to non-spatial attention remains a question for future research—a question that could be tackled on the basis of similar causal manipulations as the ones employed in this study.

On a similar note, studies have shown that the prefrontal cortex is involved in multiple control mechanisms with considerable overlap in both the involved brain structures and the mechanisms; notable examples are attention and working memory[29,30]. One idea is that working memory is also a selection mechanism, selecting behaviourally important items to facilitate manipulation and recollection of the information. We found that our electrical manipulation of the IFJ had an effect on behaviour through the attended visual evidence, but not the unattended visual evidence. An alternative interpretation could therefore be that the stimulation did not increase attention but rather increased internal manipulation and interpretation of the visual evidence through working memory. The precise distinction between attentional and working memory selection mechanisms is outside the scope of this study. We leave it for future research to better understand and distinguish these mechanisms of control.

The methodologies developed in this work and the possibility of enhancing non-spatial attention may have important implications in disorders associated with the dysregulation of top-down control. For instance, lack of success in dietary behaviour has been linked to reduced prefrontal top-down control of brain structures specialized in reward processing[31]. Failure to reduce the fear associated with traumatic experiences appears to be rooted in ineffective suppression of intrusive memories due to a lack of prefrontal top-down control over the hippocampus[32]. However, the brain–behaviour relations in these examples remain purely correlative, and whether these functions depend on top-down control remains unknown. While the effects that we observed in our study appear to be effective during the stimulation periods, it has been recently shown that repeated application of tACS can have lasting beneficial effects[33]. The possibility of selectively modulating top-down control opens the door to understanding the mechanisms of attention in higher-level cognition and developing targeted therapies in disorders associated with top-down control dysregulation.

## Methods
### Participants
The experiments conformed to the Declaration of Helsinki, and the experimental protocol was approved by the Ethics Committee of the Canton of Zurich (Kantonale Ethikkommission Zürich 2018-00659). The study tested 142 healthy young volunteers: $n$ = 20 participants took part in the fMRI study, Experiment 1 (mean age, 25.6 years; range, 21–36

years; 7 males); $n$ = 23 in the EEG study, Experiment 2 (mean age, 25.5 years; range, 19–33 years; 7 males); $n$ = 37 in the first tACS study, Experiment 3 (mean age, 25.8 years; range, 18–40 years; 22 males); $n$ = 37 in the second tACS study, Experiment 4 (mean age, 24.3 years; range, 18–35 years; 19 males); and $n$ = 37 in the third tACS study, Experiment 5 (mean age, 25.1 years; range, 19–36 years; 14 males). All participants had normal or corrected-to-normal vision. The participants were instructed about all aspects of the experiment and gave written informed consent. None of the participants suffered from any neurological or psychological disorder or took medication that interfered with participation in our study. The participants received monetary compensation of 20 CHF per hour for their participation in the experiment; in addition, they received 5 CHF per hour if they responded correctly to at least 70% of the trials.

## Stimuli

To create a behavioural task in which it is necessary to employ non-spatial attention, we created stimuli consisting of pictures and moving white dots spatially overlaid at the fovea. The visibility of these compound stimuli was dynamically modulated to follow a sinusoidal function, creating an opportunity for the visual cortices to entrain to the frequency of visual input. To make sure that the behavioural results are not contaminated by low-level confounds such as stimulus luminance or frequency spectra, we controlled the visibility of the stimuli using a phase-scrambling technique to preserve low-level image properties[34]. In brief, each image was Fourier-transformed, revealing pixel-by-pixel amplitude and phase information. A sequence of images was then generated by performing the inverse Fourier transform on a combination of the original amplitude spectrum with a modified phase spectrum. By changing the phase spectrum, we could control the recognizability of the image, while retaining identical amplitude spectra and luminance to the original image. The phase consistency could range from 0.25 (almost no picture visibility) to 0.7 (the original picture is almost fully visible). The pictures represented either indoor or outdoor scenes and were normalized to match mean luminance (SHINE toolbox, PsychToolbox). On top of the pictures, we presented 30 moving white dots; the direction of the average motion was either left or right. However, a percentage of the dots moved in a random direction; motion coherence ranged from 0.4 (almost no average direction) to 0.9 (clear average direction). The dots were shown in a circular aperture of 12°, centred at the fovea. Each dot covered 0.1° × 0.1° of the visual angle and moved at 12° per second. The complete video was sampled at the monitor's vertical refresh rate of 60 Hz. To synchronize the visual stimuli with the EEG recordings and tACS, we placed two custom-built photosensitive triggers on the sides of the monitor. This method was used in Experiment 2 to synchronize the EEG with the visual stimulation and in Experiments 3 to 5 to synchronize the visual stimulation with the electrical stimulation.

## Behavioural paradigm

The behavioural paradigm is depicted in Fig. 1a. During a trial, the participants first saw a fixation cross; afterwards, we presented a cue indicating to the participants whether they should pay attention to the motion or to the scene in the upcoming trial. Next, a sequence of four to seven compound stimuli (a scene overlaid with moving dots) was presented. After the last compound stimulus disappeared from the screen, the participants responded with a button press, taking only the last motion/scene into account. If the cue was scene, the participants were supposed to press the left arrow key if the last scenery was indoor and right if it was outdoor. If the cue was motion, the participants were supposed to press left for leftward motion and right for rightward motion. The participants had a maximum of three seconds to respond; if they failed to respond within this time, the trial was automatically incorrect. The participants were instructed to be as fast and as accurate as possible. They were rewarded with an additional 5 CHF per hour for

accuracies over 70%. Before starting the experiment, the participants took part in a training session of 64 trials starting easy and increasing in difficulty level.

In the fMRI and EEG experiments, the first 64 trials consisted of the no-attention version of the task (Supplementary Fig. 1). The participants were instructed to pay attention to the fixation cross and to press when the fixation cross changed orientation. They carried out this task with 86% and 89% accuracy for fMRI and EEG, respectively, suggesting participant engagement in this task. The information presented on the screen was nearly identical to the information in the non-spatial attention task, except words such as 'left' and 'right', which were replaced with nonsense text, and the fixation cross, which was visible at all times and occasionally rotated. This task was carried out before the participants were instructed about the non-spatial attention task, to avoid the possibility that they would pay attention to the visual stimulation other than the fixation cross (Supplementary Fig. 1).

Eye tracking measurements were acquired during all experiments in this study to control for visual engagement during task performance (EyeLink 1000 Plus, SR Research).

## fMRI (Experiment 1)

**fMRI acquisition.** The fMRI data were acquired using a 3T Philips Ingenia with the visual stimuli being presented on an LCD monitor placed behind the participant. The participants looked at the stimuli using a mirror that was attached to the head-coil. Echo planar imaging–blood-oxygen-level-dependent data were collected with a slice angle of 20° relative to the anterior–posterior commissure line, a flip angle (FA) of 85°, an echo time (TE) of 35 ms, a repetition time (TR) of 2,500 ms, 40 transversal slices (0 mm gap) and a 2.75 × 2.75 × 3.30 mm³ voxel size (field of view, 222.75 × 222.75 × 128 mm³). Participant-specific high-definition structural T1 images were acquired through a magnetization-prepared rapid gradient echo sequence with the following parameters: FA, 8°; TE, 3.6 ms; TR, 7.7 ms; voxel size, 1 × 1 × 1 mm³ (field of view, 240 × 240 × 160 mm³).

**fMRI analyses.** Analysis and preprocessing of the data were performed in FSL's Analysis Tool FEAT v.6.0.0; this included a BET brain extraction, slice timing correction, motion correction using MCFLIRT, a Gaussian spatial smoothing with a full width at half maximum of 5 mm and a high-pass temporal filtering with a cut-off of 100 s. The images were then spatially normalized using FLIRT (FMRIB's Linear Image Registration Tool), registering the low-resolution functional images to the high-resolution structural image; then, the images were warped onto the reference brain in the MNI coordinate space using FNIRT (FMRIB's Nonlinear Image Registration Tool).

First-level analysis was performed with FILM (FMRIB's Improved Linear Model) on the basis of general linear modelling with the canonical haemodynamic response function as its base function. The explanatory variables included in the analysis of the attention task performance were attention to scene, attention to motion, response to scene and response to motion. A contrast was defined for attention to scene versus attention to motion. For the passive viewing analysis, the explanatory variables were visual stimulus presentation and button presses. Group-level analysis was performed using FLAME (FMRIB's Local Analysis of Mixed Effects Tool). Contrasts were defined for attention to scene versus visual stimulus presentation and attention to motion versus visual stimulus presentation. $Z$-statistic images were thresholded at $Z$ > 2.6, and a cluster correction was applied at a threshold of $P$ < 0.05.

## EEG (Experiment 2)

**EEG acquisition and preprocessing.** EEG was acquired at 500 Hz using a high-density net (128 Channels Geodesic Sensor Net, Magstim EGI). EEG data preprocessing and analysis were performed using the Fieldtrip toolbox[35] (Donders Institute for Brain, Cognition and Behaviour, Radboud University) in MATLAB (v.R2019b, MathWorks).

Line noise was removed using a discrete Fourier transform filter. The data were re-referenced to a common average reference and epoched into 0 to 2.8 s trials to include the first four tagging cycles of each trial. We removed 179 bad trials (100 in the attention task and 79 in the no-attention task, corresponding to 4% and 6% of the trials, respectively) and 5 bad channels on the basis of visual inspection.

To quantify the neural entrainment to the visual stimulation, we computed the dWPLI[36] at 1.43 Hz between the sensor data and the imposed visibility sine wave tagging with a frequency of 1.43 Hz. We used the dWPLI because it is a phase-synchronization index that is robust to sample-size bias and spurious connectivity driven by volume conduction. This computation was performed separately for the attention and no-attention tasks, and the comparison is shown in Fig. 1f.

To localize which neural structures were entrained by the visual stimulation, source reconstruction was performed using linearly constrained minimum variance beamforming[37]. This analysis estimates the time series in each dimension for each voxel in the brain by computing spatial filters on the basis of the locations of the sensors. To reduce the dimensionality, single value decomposition was used to compute the projection with the largest variance for each voxel. To quantify the entrainment to the visual stimulation, a similar approach to the sensor-level analysis was used. The dWPLI at 1.43 Hz was computed for each voxel between the time series of the projection with the largest variance and an artificial signal of a sine wave with a frequency of 1.43 Hz corresponding to the visual stimulation. The source reconstruction and dWPLI computation were performed for the attention and no-attention tasks separately. We identified for each participant the voxels with the highest dWPLI in the attention task within a sphere of 4 cm radius centred at the frontal and occipital voxels with the highest dWPLI across participants. The time series of these voxels for one example participant is shown in Fig. 1f. To compute the delay between the visual stimulation and the neural oscillations, the source time series were band-pass filtered using a FIR filter with a cut-off of 1.43 ± 0.01 Hz. We then measured the latency of the third peak of the time series and subtracted 700 ms × 2.5 = 1.750 ms, which corresponds to the third peak of the visual stimulation. These delays are presented in Fig. 1j and were used to determine the timing of the electrical stimulation in Experiment 3 (Supplementary Fig. 8). Prior to source analysis, the data were low-pass filtered at 20 Hz using a two-pass hamming filter. The data were also high-pass filtered at 0.3 Hz with a Butterworth filter for the visualization in Fig. 1f. In this experiment, four participants were excluded due to excessive noise in the EEG recordings.

**EEG statistical analyses.** The sensor dWPLI values were compared between the attention and no-attention tasks. Cluster correction was performed by generating MCMC simulations with 5,000 permutations to determine the multiple comparison cluster correction at $P < 0.05$ on the basis of the null distribution of clusters thresholded at $P < 0.01$.

A similar approach was used for the source-level statistics. The computed dWPLIs for each voxel were compared between the attention and no-attention tasks. Cluster correction was performed by generating MCMC simulations with 5,000 permutations to determine the $P < 0.05$ threshold of the null distribution of clusters of voxels thresholded at $P < 0.01$.

### Brain stimulation (Experiments 3–5)

**tACS application.** For the application of tACS, we used a current stimulator (DC-stimulator, neuroConn) with a manual stimulation protocol controlled by MATLAB. We used two concentric ring electrodes (active electrode diameter, 2 cm; return electrode inner diameter, 7.5 cm; outer diameter, 10 cm). Following the visual sensory tagging of our behavioural paradigm, tACS was applied at a frequency of 1.43 Hz (period 0.7 s). The amplitude was determined for every participant individually with custom-written code. The maximum current used was 4 mA peak-to-peak. At the beginning and at the end of each stimulation

block, the current was ramped up and down over the first and last 10 s, respectively. As a baseline condition, we applied sham stimulation, for which we ramped up the current to its maximum amplitude over 10 s, before turning it off again. The tACS was applied continuously during the stimulation block and precisely synchronized with the visual stimuli using two photosensitive triggers attached to the monitor and custom-written code in MATLAB, which was synchronized with the computer controlling the visual input and behavioural output of our participants.

The concentric ring electrodes were placed on the scalp of the participant with the centres over the left and right IFJ. The location on the scalp that was nearest to the left and right IFJ was estimated for the test participants using a structural T1 MRI scan and neuronavigation. The average IFJ location converged in all cases into channels 117 and 128 of a EGI Geodesic 128-channel EEG cap, which was used to find the IFJ location in preparation for the tACS experiments for all participants. A topical anaesthetic (EMLA cream 5%) was used to numb the skin under the active electrodes. This procedure reduces the skin sensations induced by transcranial stimulation, which makes the stimulation more comfortable.

**Electric field predictions.** To investigate the strength of tACS exposure during the experiment, an in silico model was developed. Electromagnetic (EM) simulations were executed to predict electric field (E-field) exposures within the brain and the target region—namely, the IFJ. The EM simulations were performed using the Sim4Life (ZMT Zurich Med Tech AG) platform for computational life-science investigations, using the detailed anatomical MIDA head model[38]. The model distinguishes 37 tissue classes, of which the electric conductivities were assigned according to the IT'IS Low Frequency Database v.4.1 (ref. 39). The analysis pipeline consisted of the following steps: (1) the creation of electrode models and their placement on the skin of the MIDA (Virtual Population, IT'IS Foundation) head model, (2) identification of the anatomical target region and positioning of the electrodes, (3) execution of the EM simulations and (4) estimation of the predicted E-field distributions in the IFJ and the rest of the brain.

The target region (IFJ) in the MIDA model was identified by registering the MIDA's brain T1 images with the open-access Brainnetome Atlas[40] using FSL v.5.0 FLIRT, by importing the transformed atlas in Sim4Life and aligning it with the MIDA model. The atlas defines 246 brain areas, including left and right IFJ, which were applied as masks to the MIDA model (Supplementary Fig. 7a,b). While the stimulation target and positioning of electrodes were selected during a brain mapping procedure and defined in MNI space, in addition to coregistration of the MIDA with the Brainnetome Atlas, we developed a pipeline aimed at identifying the MNI coordinates in the MIDA brain. For this pipeline, the MIDA brain mask was first normalized to MNI space, in which the IFJ area (MNI left IFJ, −54, 12, 34 mm; MNI right IFJ, 54, 12, 34 mm) and electrode coordinates (MNI left electrode, −60, 12, 38 mm; MNI right electrode, 60, 12, 38 mm) were identified. After that, $14 \times 14 \times 14$ mm³ masks were drawn around the locations of the targets and electrodes. Finally, the normalized MIDA brain together with the new masks was coregistered with the initial MIDA brain and imported into Sim4Life. At the end of this procedure, we compared the location of the target defined in the MNI space and that of the IFJ determined with the Brainnetome Atlas, and concluded that these targets have the same positioning in the MIDA brain (Supplementary Fig. 7b). This pipeline was implemented with the SPM12 toolbox in MATLAB v.R2019a.

The electrode geometries were created in Sim4Life using the constructive geometry functionality in Sim4Life. Two cylindrical electrodes were created with radius 1 cm and were placed above the left and right IFJ, with two surrounding ring electrodes (inner radius, 4 cm; outer, 5 cm; Supplementary Fig. 7a). Two sensor boxes were placed around the central electrodes to evaluate the current and normalize the E-field distribution to the total current.

The EM simulations were executed using Sim4Life's rectilinear version of the 'Electro Ohmic Quasi-Static' finite element method solver[41]. The model geometry was discretized with a grid resolution between 0.5 and 0.75 mm (identified through a convergence analysis) with the highest refinement near the electrodes. An EM simulation was executed for each electrode, by assigning Dirichlet (voltage) boundary conditions of +1 V to the central electrode and −1 V to the ring electrode, while assigning the other electrodes to Perfect Electric Conductor. The total E-field was calculated using the superposition principle considering that the two currents are in-phase (that is, same frequency), and the focality and intensity of the stimulation were extracted on the target region.

Additionally, we ran a simulation for Experiment 5 placing the centre electrode on the Cz location of the MIDA model surrounded with the ring electrode, using the same computational parameters as for the previous simulations. The results of this model demonstrate that the E-field within the target and control regions was minimal and could not lead to activation of these areas even under 4 mA peak-to-peak stimulation: right and left IFJ (mean = 0.04, s.d. = 0.01, $P_{99}$ = 0.07), PPA (mean = 0.008, s.d. = 0.002, $P_{99}$ = 0.01) and MT/V5 (mean = 0.007, s.d. = 0.001, $P_{99}$ = 0.01; Supplementary Fig. 7e,f).

**Experiment 3: in-phase versus out-of-phase stimulation.** In Experiment 3, two stimulation conditions were used: in-phase and out-of-phase stimulation. During in-phase stimulation, the visibility of the stimulus preceded the voltage over the electrodes by 95 ± 18 ms (Supplementary Fig. 8). During out-of-phase stimulation, the timing of the voltage over the electrodes was shifted by 180° (350 ms).

The tACS experiment consisted of two sessions, each with 320 trials. In each session, either all motion trials were stimulated in-phase and all scene trials were stimulated out-of-phase or vice versa. For half of the participants, pseudo-randomly chosen, the first session consisted of the in-phase stimulation condition for trials in which scene was cued and out-of-phase stimulation for trials in which motion was cued. For these participants, the second session consisted of in-phase stimulation for motion trials and out-of-phase stimulation for scene trials. For the second half of the participants, the order of the stimulation conditions was reversed. A single session was divided into four blocks. Stimulation was turned on only in blocks two and three. Blocks one and four consisted of sham stimulation. In half of the trials of blocks one and four, the participant was cued after the visual stimulus had disappeared, thus making the participant pay attention to both the scene and the motion during stimulus presentation. Participants with discrimination performance <55% in block one (suggesting nearly random choice selection and therefore poor engagement) were excluded from the data analyses, resulting in the exclusion of four participants.

**Experiment 4: tACS phase-dependent effects.** In the phase-dependent effects tACS experiment, we used six stimulation delays spaced out evenly over the period of a single visual stimulus period. To maximize statistical power over the six different conditions (which were in turn divided into the four sensory evidence levels used in Experiment 3), the experimental session consisted of the continuous application of tACS. The stimulation delays were pseudo-randomly assigned on a trial-to-trial basis. This experiment thus allows us to study the relationship between the ongoing phase of the tACS stimulation and the presentation of the visual stimulus. Details regarding the statistical analyses are described in 'Behavioural analysis and statistics'. Participants with discrimination performance <55% in block one (suggesting nearly random choice selection and therefore poor engagement) were excluded from the data analyses, resulting in the exclusion of five participants.

**Experiment 5: control stimulation location.** We placed the centre of the electrodes over the Cz location of the 10–20 EEG coordinate system, therefore stimulating the motor cortex. All other experimental parameters were equal to those of Experiment 4. Participants with discrimination performance <55% in block one (suggesting nearly random choice selection and therefore poor engagement) were excluded from the data analyses, resulting in the exclusion of three participants.

**Eye tracking**

Eye tracking (EyeLink 1000 Plus) was used to check the participants' eye movement during stimulus presentation. A chin rest was used to keep the distance between the participants and the screen constant (55 cm). Microsaccade data were extracted and analysed using the widely adopted approach described by Engbert and Kleigl[42]. We focused on the combination of saccades and microsaccades (saccades <1° of the visual angle) occurring within the first four tagging cycles of each trial (the first 2.8 s of stimulus presentation).

**Behavioural analysis and statistics**

**Mixed-effects model of sensory discrimination behaviour.** A logistic mixed model was implemented to investigate the effect of stimulation (in-phase or out-of-phase) on the participant's sensory discrimination as a function of both the cued and the distractor sensory evidence. In trials in which the participant was cued to pay attention to motion, the motion evidence is the main explanatory variable, while scene evidence is a distractor that should be ignored and vice versa. The log-odds for making the left or right decision is given by

$$\bar{\beta} = \beta_0 + \beta_1 \times \text{motion} + \beta_2 \times \text{scene} + \beta_3 \times \text{stimulation}$$
$$+ \beta_4 \times \text{motion} \times \text{stimulation} + \beta_5 \times \text{scene} \times \text{stimulation}, \quad (1)$$

where the probability of selecting 'right' and explaining the participant's ($p$) response $y_{i,p} \in \{0, 1\}$ (with $y = 0$ indicating 'left' and $y = 1$ indicating 'right') in trial $i$ is given by

$$\theta_{i,p} = 1/\left(1 + e^{-\bar{\beta}}\right)$$
$$y_{i,p} \sim \text{Bernoulli}(\theta_{i,p}). \quad (2)$$

A positive interaction effect between stimulation and sensory evidence (motion or scene) indicates that the corresponding sensory information influences the participant's behaviour more strongly in the in-phase stimulation condition than in out-of-phase.

The model has six parameters that need to be estimated; we placed uninformative priors between sensible limits on all parameters as follows:

$$\beta_{0-5} \sim \text{Normal}(0, 0.001) \quad (3)$$

**Dynamic evolution analyses of stimulation effects.** To study how the stimulation influenced task performance over time, a moving-window analysis of tACS influences on behaviour was performed with a window length of 90 trials. For each window, a logistic mixed-effects model similar to the one described above was fitted to the behavioural choice data. In the corresponding figures, we report the standardized interaction of evidence × stimulation with the error denoting ±1 s.d. The interaction effects were cluster corrected at $P < 0.05$ by constructing a null distribution of cluster sizes, on the basis of shuffling the labels of the stimulation phase data within participants.

**Computational model.** Our brain stimulation protocol appears to induce attentional influences in sensory discrimination performance. However, these results do not clarify whether these behavioural modulations are indeed specific to boosting the perception of sensory evidence. A way to clarify this would be to apply tACS during neuroimaging. However, due to technical and safety aspects, we were not able to apply current intensities above 2 mA peak-to-peak, while in our behavioural studies we applied currents of up to 4 mA peak-to-peak. It

was therefore not possible to combine tACS and fMRI with the protocol developed here. Nevertheless, this question can be tackled with the use of computational models.

We analysed the influence of tACS on the discriminability of the cued sensory feature with a prominent mathematical model of two-alternative decisions, the drift-diffusion model, which incorporates both observed choices and reaction times (RTs) to decompose the decision process into distinct latent variables corresponding to distinct aspects of the choice process: (1) the efficiency of sensory evidence accumulation, known as the drift rate ($\delta$); (2) any bias in the choice process ($\beta$); (3) the amount of evidence required to make a decision, known as the decision threshold ($\alpha$); and (4) the delay in the onset of evidence accumulation, the non-decision time ($\tau$).

The decision-making model implemented here is based on a simple one-dimensional Wiener process: a dynamic system where the state of evidence $X(t)$ at time $t$ evolves via the stochastic equation $\frac{dX(t)}{dt} \sim \text{Normal}(\delta, \sigma^2)$ where $\delta$ represents the quality of information processing defined as $\delta = kE$, where $E$ represents the sensory evidence level (that is, the stimulus visibility in our task) and $k$ a variable that linearly scales the evidence. For the initial conditions, where $\beta$ represents an initial bias in the process, it is assumed that the system makes a decision $\zeta$ (left or right) at time $t$ whenever $X(t) \geq \alpha$ (right) or $X(t) < 0$ (left). In addition, we accounted for visual processing and corticomuscular response delays via the non-decision time parameter $\tau$ (the RT in each trial is defined as $RT = t + \tau$). The goal is to find the Wiener distribution, Wiener($\delta, \alpha, \tau, \beta$), that best explains the distribution of empirical choices $y(\zeta, RT)$. To this end, we implemented a hierarchical Bayesian model where each individual data point $y_{i,p}(\zeta, RT)$ follows a Wiener distribution

$$y_{i,p} \sim \text{Wiener}(\delta, \alpha, \tau, \beta), \tag{4}$$

with indices $p$ for participants ($p = 1, \dots, N_{\text{participants}}$) and $i$ for trials ($i = 1, \dots, N_{\text{trials}}$).

Given that in our study we used a hierarchical Bayesian data analysis framework, this allows the convenient possibility of studying the effects of a given tACS stimulation condition (for example, in-phase stimulation) on a latent variable during a baseline condition (for example, the drift-rate modulator $k$ during out-of-phase stimulation or baseline trials). We thus studied the (potential) relative change of a given latent variable $\theta \in \{k, \alpha, \tau, \beta\}$ as follows:

$$\theta_{p,i} = \theta_{\text{base},p} + \beta_p^\theta \times D_i, \tag{5}$$

where $D \in \{1, 0\}$ denotes whether the modulator condition (for example, in-phase stimulation in our example) was present ($D = 1$) or not ($D = 0$) in each trial $i$. The subscript $p$ denotes that the effect is participant-specific, which is modelled as a random-effects factor under the assumption that it is drawn from population distributions $\theta_{\text{base},p} \sim N(\theta_{\text{base}}, \sigma_{\text{base}})$ and $\beta_p^\theta \sim N(\beta^\theta, \sigma^\theta)$ where $\theta_{\text{base}}, \beta^\theta$ and $\sigma_{\text{base}}, \sigma^\theta$ determine the mean and the standard deviation of the population distributions, respectively.

The model has eight parameters that need to be estimated; we placed uninformative priors between sensible limits on all parameters as follows:

$$
\begin{aligned}
k_{\text{base}} &\sim \text{Uniform}(-5, 5) \\
\alpha_{\text{base}} &\sim \text{Uniform}(0.001, 10) \\
\tau_{\text{base}} &\sim \text{Uniform}(0.001, 0.5) \\
\beta_{\text{base}} &\sim \text{Uniform}(0.0001, 0.9999) \\
\beta^{k,\alpha,\tau,\beta} &\sim \text{Uniform}(-1, 1)
\end{aligned}
\tag{6}
$$

**Sinusoidal model (Experiments 4 and 5).** The aim of Experiments 4 and 5 was to study whether the ongoing tACS phase relative to a single stimulus presentation modulates non-spatial attention behaviour. We synchronized the ongoing tACS peak at six equally spaced phase delays over one full stimulation period (Fig. 3a). To study the influence of the delays in a parsimonious parametric model, we first performed a separate logistic regression for each participant and each stimulation delay condition as follows:

$$
\begin{aligned}
\theta_i &= 1/\left(1 + e^{-(\beta_{p,d} + E_i \times \delta_{p,d})}\right) \\
y_i &\sim \text{Bernoulli}(\theta_i),
\end{aligned}
\tag{7}
$$

where $y_i \in \{0, 1\}$ denotes the trial-to-trial choices in each trial $i$ as a function of $E_i$, which denotes the amount of motion evidence in the trials in which motion was cued and the amount of scene evidence in the trials where scene was cued. $\beta_{p,d}$ is a participant ($p$) and stimulation delay ($d$) specific bias term, and $\delta_{p,d}$ corresponds to a participant and stimulation delay specific slope. We next fit a sinusoidal function through the slope parameters of the logistic regression as a function of stimulation delay:

$$
\begin{aligned}
\mu_{p,d}(\tau_d) &= \beta_p + A_p \sin\left(\frac{2\pi\tau_{p,d}}{6} + \phi_p\right) \\
\delta_{p,d} &\sim N(\mu_{p,d}, \sigma_d),
\end{aligned}
\tag{8}
$$

where $\delta_{p,d}$ is the population distribution of the participant-specific psychometric slopes for each stimulation delay ($d$) obtained in equation (7), and $\tau_{p,d}$ is the timing of the different tACS phase delays. $\beta_p$ represents the participant-specific offset of the sinusoidal function with amplitude $A_p$. Parameter $\phi_p$ determines the phase shift, which was parameterized as a von Misses distribution

$$\phi_p \sim \frac{\exp(\kappa \cos(x - \phi))}{2\pi I_0(\kappa)}, \tag{9}$$

initialized with a flat prior (that is, $\kappa = 0$), where $I_0$ is the modified Bessel function of the first kind of order 0.

Here it is important to emphasize that the key parameter determining a tACS phase-delay modulation is the population-level estimate of the sinusoidal amplitude, which is estimated departing from an exponential prior distribution

$$A_p \sim \lambda e^{-\lambda A} > 0, \tag{10}$$

with a conservative prior by setting $\lambda = 4$. However, we found that our results are largely insensitive to the selection of this prior. Note that this conservative prior promotes smaller amplitudes, as psychometric slopes larger than 1 are unlikely. Given that this parameter is by definition positive, the significance of the expected amplitude at the population level $\mathbb{E}[A]$ was determined by comparing this value to a null distribution of expected amplitude values $\mathbb{E}[A]_{\text{rand}}$, on the basis of shuffling the labels of the stimulation phase data within participants and repeating the procedure described in equations (6)–(8) 5,000 times to estimate each $\mathbb{E}[A]_{\text{rand}}$. To compare the effects of tACS across conditions, we obtained a standardized estimate of the amplitude modulation effect $z_A$. Assuming that the null distribution of expected amplitude values $\mathbb{E}[A]_{\text{rand}}$ approximates a normal distribution, we defined the standardized estimate of the amplitude modulation effect $z_A$ as

$$z_A = \sqrt{2}\,\text{erf}^{-1}(2P - 1), \tag{11}$$

where $P$ is the proportion of samples of the null distribution smaller than $\mathbb{E}[A]$, and $\text{erf}^{-1}(x)$ is the inverse of the error function $\text{erf}(x)$:

$$\text{erf}(x) = \frac{2}{\sqrt{\pi}} \int_0^x e^{-t^2} dt. \tag{12}$$

We also performed a Bayes factor analysis to investigate the support for the tACS-induced effects favouring or disfavouring the null, where the null was calibrated using the empirical null. Note that on the basis of the resulting null distributions obtained from our data (which are, moreover, distinct for each sensory modality; Fig. 3), it would be incorrect to assume that the null corresponds to $A = 0$. Thus, we more appropriately calibrated the null for the Bayes factor analyses on the basis of the empirical null. Our hierarchical Bayesian analyses give us direct information of the mean $\mu_A$ and standard deviation $\sigma_A$ of the sinusoidal amplitude modulation $A$ for the empirical data at the population level (which we assumed in our model described above to be normally distributed—that is, $A_p \sim N(\mu_A, \sigma_A)$), as well as for the corresponding inferred latent variables for each point of the null distribution. Therefore, we first computed a $d'$ estimate of the amplitude modulation effect using the posterior estimates of the mean and variance of the mean of the empirical data and the corresponding values at the expected value of the null:

$$d' = \frac{\mu_A - \mu_{A_{\text{rand}}}}{\sqrt{\sigma_A^2 + \sigma_{A_{\text{rand}}}^2}}. \tag{13}$$

The Bayes factor was computed on the basis of $d'$ assumed to be 0 under the null hypothesis and Cauchy with scale parameter $s = 2/\sqrt{2}$ under the alternative. That is, one would expect $d' = 0$ if the stimulation does not induce sensory performance modulations different from what one would expect by chance.

The model has three parameters that need to be estimated; we placed uninformative priors between sensible limits on all parameters as follows:

$$\begin{aligned} \beta &\sim \text{Normal}(1, 0.01) \\ A &\sim \text{Exponential}(4) \\ \phi &\sim \text{Uniform}(0, 2\pi) \end{aligned} \tag{14}$$

**Statistical inference.** All mixed-effects models in this study had varying subject-specific latent variables unless otherwise specified in each model description. Posterior inference of the parameters in the hierarchical models was performed via the Gibbs sampler using the MCMC technique implemented in JAGS, assuming flat priors for all population-level parameters (unless otherwise specified). For each model, a total of 100,000 samples were drawn from an initial burn-in step, and subsequently, a total of 100,000 new samples were drawn with three chains (each chain was derived on the basis of a different random number generator engine, and each with a different seed). We applied a thinning of 100 to this final sample, resulting in a final set of 1,000 samples for each parameter. We conducted Gelman–Rubin tests for each parameter to confirm the convergence of the chains. All latent variables in our Bayesian models had $\hat{R} < 1.05$, which suggests that all three chains converged to a target posterior distribution. We checked via visual inspection that the posterior population-level distributions of the final MCMC chains converged to our assumed parameterizations. For all random effects reported here, the reported value corresponds to the mean of the standardized posterior distribution, and the 'P values' reported for these regressions are not frequentist P values but instead directly quantify the probability of the reported effect differing from zero ($P_{\text{MCMC}}$). They were computed using the posterior population distributions estimated for each parameter and represent the portion of the density function that lies above/below 0 (depending on the direction of the effect). The standardized effects of the hierarchical mixed-effects models reported in the main text were obtained by dividing the expected value of the corresponding posterior $\beta$ estimate by its standard deviation.

## Reporting summary

Further information on research design is available in the Nature Portfolio Reporting Summary linked to this article.

## Data availability

The data that support the findings of this study are available on request from the corresponding authors. The data are not publicly available due to information that could compromise the privacy of the research participants. Minimal source data to replicate all figures in the Article can be found on the Open Science Framework at https://doi.org/10.17605/OSF.IO/H7EKC.

## Code availability

All code needed to replicate the results presented in this study has been made available on the Open Science Framework at https://doi.org/10.17605/OSF.IO/H7EKC.

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

## Acknowledgements

This work was supported by a European Research Council starting grant (ENTRAINER) to R.P., an ETH Grant (ETH-25 18-2) to R.P. and a ZNZ PhD grant to J.B. and R.P. This project has received funding from the European Research Council under the European Union's Horizon 2020 research and innovation programme (grant agreement no. 758604). This work was supported by a grant awarded to R.P. and L.I. from the Swiss National Science Foundation (SNF 197766). The funders had no role in study design, data collection and analysis, decision to publish or preparation of the manuscript. We also thank J. Qiuhan for helping pilot the experimental paradigm; I. Gezginer for helping collect the EEG data; M. Quiniou, H. van der Pol and S. Warma for helping collect the tACS data; and M. Compaijen for proof-reading the manuscript.

## Author contributions

J.B. wrote experimental code, collected fMRI and tACS data, performed data analysis and implemented computational models. J.A.H. wrote experimental code and performed EEG data analysis. V.B., A.M.C. and E.N. performed computational modelling of electric fields. F.G.P. collected EEG data, performed EEG analysis and troubleshot the experimental set-ups. J.B., J.A.H., V.B., M.G. and R.P. contributed to the conceptual development of the experimental paradigm. All authors discussed the results and wrote the paper. R.P. and L.I. raised the funding.

## Funding

## Competing interests

The authors declare no competing interests.

## Additional information

**Correspondence and requests for materials** should be addressed to Jeroen Brus or Rafael Polanía.

# Reporting Summary

## Statistics

For all statistical analyses, confirm that the following items are present in the figure legend, table legend, main text, or Methods section.

| n/a | Confirmed | |
|---|---|---|
| ☐ | ☒ | The exact sample size (*n*) for each experimental group/condition, given as a discrete number and unit of measurement |
| ☐ | ☒ | A statement on whether measurements were taken from distinct samples or whether the same sample was measured repeatedly |
| ☐ | ☒ | The statistical test(s) used AND whether they are one- or two-sided<br>*Only common tests should be described solely by name; describe more complex techniques in the Methods section.* |
| ☐ | ☒ | A description of all covariates tested |
| ☐ | ☒ | A description of any assumptions or corrections, such as tests of normality and adjustment for multiple comparisons |
| ☐ | ☒ | A full description of the statistical parameters including central tendency (e.g. means) or other basic estimates (e.g. regression coefficient) AND variation (e.g. standard deviation) or associated estimates of uncertainty (e.g. confidence intervals) |
| ☐ | ☒ | For null hypothesis testing, the test statistic (e.g. *F*, *t*, *r*) with confidence intervals, effect sizes, degrees of freedom and *P* value noted<br>*Give P values as exact values whenever suitable.* |
| ☐ | ☒ | For Bayesian analysis, information on the choice of priors and Markov chain Monte Carlo settings |
| ☐ | ☒ | For hierarchical and complex designs, identification of the appropriate level for tests and full reporting of outcomes |
| ☒ | ☐ | Estimates of effect sizes (e.g. Cohen's *d*, Pearson's *r*), indicating how they were calculated |

*Our web collection on statistics for biologists contains articles on many of the points above.*

## Software and code

Policy information about availability of computer code

| Data collection | Matlab R2019a, Psychtoolbox-3.0.14, SHINE toolbox |
|---|---|
| Data analysis | Matlab R2019a, R3.6.3, JAGS 4.3.0, Fsl v6.0.1&v5.0, FEAT v6.0.0, MCFLIRT, FLIRT, FNIRT, FILM, FLAME, Fieldtrip 2019.11.26, SPM12 and Sim4Life |

For manuscripts utilizing custom algorithms or software that are central to the research but not yet described in published literature, software must be made available to editors and reviewers. We strongly encourage code deposition in a community repository (e.g. GitHub). See the Nature Portfolio guidelines for submitting code & software for further information.

## Data

Policy information about availability of data

All manuscripts must include a data availability statement. This statement should provide the following information, where applicable:

- Accession codes, unique identifiers, or web links for publicly available datasets
- A description of any restrictions on data availability
- For clinical datasets or third party data, please ensure that the statement adheres to our policy

Source data for all figures can be found at the Open Science Foundation at: XX. Full data that support the findings of this study are available on request from the corresponding authors. The data are not publicly available due to information that could compromise the privacy of the participants. The study also made use of the Brainnetome Atlas https://atlas.brainnetome.org/ and the automated anatomical labelling atlas (AAL3v1) https://www.oxcns.org/aal3.html.

## Human research participants

Policy information about studies involving human research participants and Sex and Gender in Research.

| | |
|---|---|
| Reporting on sex and gender | This study tested 142 young volunteers of which 69 were male and 73 were female. Their sex was determined via self-reporting but was not relevant to any analysis we performeded. |
| Population characteristics | All participants were healthy and had normal or corrected to normal eye-sight. None of the participants suffered from any neurological or psychological disorder or took medication that interfered with participation in our study. In the fMRI experiment the mean age is 25.6, EEG 25.5, first tACS 25.8, second tACS 24.3 and third tACS 25.1. |
| Recruitment | Participants were recruited at the online participant recruitment platforms of the ETH Zürich and the UZH. This recruitment procedure biases towards young and highly educated participants, the impact of this recruitment bias on the results is hard to estimate. |
| Ethics oversight | Ethics Committee of the Canton of Zürich |

Note that full information on the approval of the study protocol must also be provided in the manuscript.

# Field-specific reporting

Please select the one below that is the best fit for your research. If you are not sure, read the appropriate sections before making your selection.

☐ Life sciences    ☒ Behavioural & social sciences    ☐ Ecological, evolutionary & environmental sciences

For a reference copy of the document with all sections, see nature.com/documents/nr-reporting-summary-flat.pdf

# Behavioural & social sciences study design

All studies must disclose on these points even when the disclosure is negative.

| | |
|---|---|
| Study description | We have acquired quantitative experimental data using fMRI, EEG and behavioral studies, including tACS. |
| Research sample | The study tested healthy young volunteers (n=142, age 18-40) recruited through the UZH and ETH web page available for recruitment of participants. The sample is representative of young healthy individuals studying at the ETH Zürich and the UZH. This follows common practices for participant recruitment at the ETH. |
| Sampling strategy | Sample size was determined based on previous studies using similar stimuli and tasks (Zanto 2011). Sampling procedure was based on voluntary response. |
| Data collection | The experiment was implemented in Matlab with the use of Psychtoolbox. Participants eye movements were recorded and depending on the experiment fMRI, EEG or tACS techniques were used. Nobody was present during the experiment except for the participant and experimenter. The reseacher was blinded to the stimulation condition. |
| Timing | Data was collected from March 2019 to September 2022. |
| Data exclusions | Four participants were excluded due to excessive noise in the EEG recordings. Participants with discrimination performance < 55% in block one (suggesting nearly random choice selection and therefore poor engagement) were excluded from the data analyses, resulting in the exclusion of 12 participants in experiments 3,4&5. |
| Non-participation | No participants dropped out or declined participation. |
| Randomization | Participants were not allocated to experimental groups. |

# Reporting for specific materials, systems and methods

We require information from authors about some types of materials, experimental systems and methods used in many studies. Here, indicate whether each material, system or method listed is relevant to your study. If you are not sure if a list item applies to your research, read the appropriate section before selecting a response.

## Materials & experimental systems

| n/a | Involved in the study |
|-----|----------------------|
| ☒ | ☐ Antibodies |
| ☒ | ☐ Eukaryotic cell lines |
| ☒ | ☐ Palaeontology and archaeology |
| ☒ | ☐ Animals and other organisms |
| ☒ | ☐ Clinical data |
| ☒ | ☐ Dual use research of concern |

## Methods

| n/a | Involved in the study |
|-----|----------------------|
| ☒ | ☐ ChIP-seq |
| ☒ | ☐ Flow cytometry |
| ☐ | ☒ MRI-based neuroimaging |

# Magnetic resonance imaging

## Experimental design

| | |
|---|---|
| Design type | Event related design |
| Design specifications | We tested 20 participants who each performed 192 trials divided over 6 blocks. Each trial took about 7 seconds with an intertial interval between 1 and 3 seconds. |
| Behavioral performance measures | Correct button presses and reaction times were recorded. Performance over 55% was used as a criterion for sufficient task performance. |

## Acquisition

| | |
|---|---|
| Imaging type(s) | functional and structural |
| Field strength | 3 Tesla |
| Sequence & imaging parameters | Pulse sequence type was gradient echo with EPI imaging, FOV = 222.75x128, flip angle 85 degrees, slice angle of 20 degrees relative to the anterior-posterior commissure line. |
| Area of acquisition | Whole brain |
| Diffusion MRI | ☐ Used    ☒ Not used |

## Preprocessing

| | |
|---|---|
| Preprocessing software | Analysis and pre-processing of the data was performed in FSL's Analsysis Tool FEAT v6.0.0, this included a BET brain extraction, slice timing correction, motion correction using MCFLIRT, a Gaussian spatial smoothing with a full width at half maximum of 5 mm, and a high pass temporal filtering with a cut-off of 100s |
| Normalization | Images were spatially normalized using FLIRT |
| Normalization template | The functional images were normalized to the high resolution structural image and then using FNIRT were warped onto the reference brain in MNI coordinate space. |
| Noise and artifact removal | There was a slice timing correction and a motion correcion using MCFLIRT. |
| Volume censoring | There was no volume censoring |

## Statistical modeling & inference

| | |
|---|---|
| Model type and settings | First level analysis was performed with FILM based on general linear modelling with the canonical hemodynamic response function. Group-level analysis was performed using FMRIB's Local Analysis of Mixed Effects Tool. |
| Effect(s) tested | Contrasts were defined for attention to scene vs visual stimulus presentation and attention to motion vs visual stimulus presentation. |
| Specify type of analysis: | ☒ Whole brain    ☐ ROI-based    ☐ Both |
| Statistic type for inference (See Eklund et al. 2016) | Cluster wise omparison with a cluster correction at a threshold of P < 0.05 |
| Correction | - |

## Models & analysis

| n/a | Involved in the study |
|-----|----------------------|
| ☒ | ☐ Functional and/or effective connectivity |
| ☒ | ☐ Graph analysis |
| ☒ | ☐ Multivariate modeling or predictive analysis |

