## [Peer Review File · Nature Human Behaviour]

Peer Review Information

Journal: Nature Human Behaviour

Manuscript Title: Causal phase-dependent control of non-spatial attention in human prefrontal cortex

Corresponding author name(s): Jeroen Brus and Rafael Polanía

Reviewer Comments & Decisions:

Decision Letter, initial version:

26th April 2023

Dear Prof Polania,

Thank you once again for your manuscript, entitled "Causal phase-dependent control of feature-based attention in human prefrontal cortex," and for your patience during the peer review process.

Your manuscript has now been evaluated by 3 reviewers, whose comments are included at the end of this letter. Although the reviewers find your work to be of interest, they also raise some important concerns. We are interested in the possibility of publishing your study in Nature Human Behaviour, but would like to consider your response to these concerns in the form of a revised manuscript before we make a decision on publication.

To guide the scope of the revisions, the editors discuss the referee reports in detail within the team, including with the chief editor, with a view to (1) identifying key priorities that should be addressed in revision and (2) overruling referee requests that are deemed beyond the scope of the current study. We hope that you will find the prioritized set of referee points to be useful when revising your study. Please do not hesitate to get in touch if you would like to discuss these issues further.

First, we ask that you address two important concerns by Reviewer #2 regarding the lack of a control site in Exp 3 and the difference in sample size in Exp 5 (comments 4 and 5). Please [1] perform an additional control experiment that includes better control for the tactile stimulation induced by tACS and a proper control site. Please [2] use Bayes Factors to quantify support for the null hypothesis in Experiment 5. If the evidence in support of the null is inconclusive (see <https://www.ejwagenmakers.com/2015/BayesianAnalysisEnclopedia.pdf>), please increase the sample size to be the same as for Experiment 3 and use Bayesian statistics to quantify support for the null. (If you decide to continue using NHST for Experiment 5 after adding more participants, you will need to use sequential analyses to control for Type I error, which is inflated when repeated analyses are performed in NHST). We also ask that you [3] answer the questions regarding the specificity of the effects in IFJ given the spatial resolution of the methods used in the study (comment 3 by Reviewer #2 and comment 2 by Reviewer #3). Finally, please [4] clarify the definition of feature-based attention used in the manuscript (comment 1 by Reviewer #1, comment 1 by Reviewer #2).

In sum, we invite you to revise your manuscript taking into account all reviewer and editor comments. We are committed to providing a fair and constructive peer-review process. Do not hesitate to contact us if there are specific requests from the reviewers that you believe are technically impossible or unlikely to yield a meaningful outcome.

We hope to receive your revised manuscript within two months. I would be grateful if you could contact us as soon as possible if you foresee difficulties with meeting this target resubmission date.

- Include a “Response to the editors and reviewers” document detailing, point-by-point, how you addressed each editor and referee comment. If no action was taken to address a point, you must provide a compelling argument. When formatting this document, please respond to each reviewer comment individually, including the full text of the reviewer comment verbatim followed by your response to the individual point. This response will be used by the editors to evaluate your revision and sent back to the reviewers along with the revised manuscript.
- Highlight all changes made to your manuscript or provide us with a version that tracks changes.

[REDACTED]

We look forward to seeing the revised manuscript and thank you for the opportunity to review your work. Please do not hesitate to contact me if you have any questions or would like to discuss these revisions further.

Sincerely,

Giacomo Ariani
Editor
Nature Human Behaviour

Reviewer expertise:

Reviewer #1: Rhythmic feature-based attention, fMRI, EEG, TMS

Reviewer #2: Rhythmic feature-based attention, EEG, TMS

Reviewer #3: Rhythmic feature-based attention, EEG

REVIEWER COMMENTS:

Reviewer #1:

Remarks to the Author:

Review of Causal phase-dependent control of feature-based attention in human prefrontal cortex by Brus and colleagues.

The authors present a new approach that combines a behavioral task, fMRI, EEG and non-invasive tACS brain stimulation to predict and experimentally control the excitability states of the inferior frontal junction (IFJ) during a non-spatial, ‘feature-based’ attention task.

The show that applying electrical stimulation (tACS) to IFJ led to changes in performance, indicating a causal role of the phase-alignment in this region of prefrontal cortex. This effect was further replicated in another experiment that did not require steady-state tagging of the IFJ through visual stimulation, but allowed for the study of systematic phase delays between tACS and the visual stimulus to be processed.

While the results for visual scenes were not significant in the original experiment with fixed in- and out-of-phase stimulations, the subsequent experiments showed significant modulation of attention to scenes when using a broader variety of stimulation timings, possibly reflecting the more complex pattern of sensory features in scenes.

The manuscript is well-written and carefully prepared. In the sum of all its parts, it draws a detailed picture of the causal involvement of IFJ in feature-based top-down attention. The authors invested a lot of effort in optimizing their stimulation paradigms and combining different imaging and

electrophysiological measures within this one study. The conclusions that are drawn from the experiments are all well-supported by the data and go far beyond our current knowledge of this special region in PFC.

Despite all these positive aspects, I think there is some general confusion about feature- vs. object-based attention. This confusion is already present in the Introduction of the paper, but continues also in the Methods, and in the interpretation of the results. Given the rather complex task the authors chose to study IFJ's involvement in top-down attention, I think it would help to set a better theoretical foundation of the various aspects of attentional control in which IFJ might be involved. Particularly, the task combines a traditional feature-based attention task (in form of the coherent motion stimuli) with an object-based attention task (in form of the scene classification task). The key difference is that scenes are complex visual stimuli that are made up of many different features, often even of different feature dimensions. Although these are rather different tasks from an attentional point of view, the combination is justified in the sense that a) IFJ is involved in both feature- and object-based attention, and b) they are both non-spatial forms of attention, which is in stark contrast to classical spatial attention paradigms, in which IFJ is less involved. Therefore, the emphasis of the study seems to be more on the specific aspect of studying non-spatial attention – in contrast to spatial attention paradigms that are supposed to be governed by a dorsal attention system implying FEF and PPC instead. Actually, the authors might find it helpful to have a look into a recent review by Bedini & Baldauf, 2021 EJM, who described in much detail the division of labor between IFJ and FEF (both within PFC) in governing non-spatial (i.e. feature- and object-based) and spatial attention, respectively.

Also in the Introduction, the authors motivate their study by referring to previous research that has shown that attentional performance fluctuates over time and that control structures might provide their feedback signals to visual cortex rhythmically. This is true, however, I think all the experimental evidence for this notion stem from paradigms that used spatial attention tasks. Therefore, the current study might be the first one actually to show that such rhythmic attentional sampling occurs also for non-spatial, feature-based attention.

The Discussion section of the manuscript is rather short and ends somewhat abruptly, leaving some questions and implications of the study open. For example, what are the underlying anatomical and functional connectivities of IFJ, given its important (and here even causal) role in the mentioned attention tasks? In the end, the function of a brain area is determined by its intrinsic and extrinsic connectivity fingerprints. Some first indications for tight anatomical connections between IFJ and feature-/object-processing visual areas in IT cortex (as assessed with diffusion imaging for example) have already been provided by Baldauf & Desimone, 2014, showing that the more ventral parts of lateral PFC have tight connections with scene-processing areas like PPA. Further, Bedini and colleagues (Bedini et al., 2021 Cognitive Processing), presented recently some novel data on IFJ's anatomical connectome (based on probabilistic diffusion imaging tracking), showing its preferential connectivity with the ventral visual pathway (which is the one crucial for feature-based attention), particular in contrast to spatially organized control sites like FEF. They concluded from their tractography data that also in PFC there is a subdivision in a dorsal and ventral part responsible for spatial vs non-spatial top-down signals, respectively. On the functional connectivity side, IFJ has been shown to have systematic fMRI co-activations (as assessed with the help of meta-analytic connectivity modeling, MACM) with the ventral visual pathway containing areas that process high-level feature and object information (Bedini et al., bioRxiv/Brain, Structure and Function, 2023). Furthermore, an analysis of IFJ's functional connectivity profile in resting state MEG has shown that it is tightly connected with many important high-level visual areas processing feature information, especially in the delta, beta, and gamma frequencies (see Soyuhos & Baldauf, EJM 2023).

Also in the discussion section, I was missing a paragraph on the functional specificity of IFJ's involvement in top-down attentional guidance. In particular, it has been proposed that many prefrontal regions, including IFJ, might belong to a more general 'multiple-demand system' (Duncan, 2010; Fedorenko et al., 2012). Here the reported activations seem much more focal and specific to IFJ itself. I think, a very important aspect that might explain the specific activation of IFJ in the current tasks is the carefully designed functional contrast in Fig. 1b, which used (as in Baldauf & Desimone, 2014) difficulty matched contrasts in fMRI. The matching of task difficulty might have helped in avoiding more wide-spread, and task un-specific activations in the contrasts 'Motion/Scene>no attention'.

I couldn't find anywhere the exact coordinates of the peak activations of IFJ. It would be helpful for future studies to provide the x/y/z locations. Do they fit well with the accurate location of IFJ, as for example, recently identified by Bedini and colleagues using activation likelihood estimations (ALE) across a large set of fMRI localizer studies (Bedini et al., bioRxiv/Brain, Structure and Function, 2023)?

signed, Daniel Baldauf

Reviewer #2:

Remarks to the Author:

SUMMARY

The aim of this study is to show that exogenously modulating prefrontal excitability leads to phase-specific modulations in feature-based attention. Previous work linking prefrontal excitability and oscillations to feature-based attention have largely been correlative. This manuscript comprises five separate studies using either fMRI, EEG or tACS, and in addition modelling. In the main task throughout, participants see sequences of visual stimuli in which moving dots are superimposed onto pictures of scenes. A cue starting each trial instructs whether to discriminate dot direction or scene category. Then participants respond to the last presentation in each sequence. In a separate 'no attention' control task, always performed first, participants respond to fixation cross changes. In Experiments 1- 3 the visual stimuli fluctuate in visibility at 1.4 Hz to allow measuring for entrainment.

Experiment 1 finds as would be expected that motion or scene-specialised cortical areas are activated most when reporting to that type of information, and a general attentional network including the IFJ is important for both, compared to the control task.

Experiment 2 records 128-channel EEG during this task. Many electrodes show a signal entrained to the frequency of the visibility fluctuation (using a measure of phase synchronisation, dWPLI). A source analysis reveals a cluster in the EEG data estimated to be around the IFJ to correlate with the visibility fluctuation more during attention than the no-attention blocks. The phase lag of this frontal cluster was delayed compared to those from occipital electrodes. An eye tracking analysis shows that microsaccades cannot explain differences in brain activity between different conditions.

Experiment 3 then applies frontal tACS with one electrode over the IFJ and a concentric ring of return electrode around it. In different sessions, participants receive tACS either in phase or out of phase with the visual stimulus fluctuations. In phase stimulation was only better than out of phase stimulation for the motion stimuli. The tACS effect is found to disappear after tACS ends is found in a drift diffusion model to specifically affect the rate of sensory evidence accumulation.

Experiment 4 drops the visual fluctuations and explores the effect of 6 different tACS phases using either just motion stimuli or just scenes, finding phase effects on both.

Experiment 5 repeats experiment 4 by stimulating a vertex control site but does not find behavioural effects.

MAJOR COMMENTS

This is in many ways an impressive manuscript using several techniques, but raises several queries which detract from the conclusions.

More support is needed to explain why these effects are taken as showing specifically feature-based attention. It seems that the lack of distraction by the other task (motion versus scene) is taken to indicate that participants are orienting attention to the cued feature. But participants could be enhancing the features of both stimulus sets during attention blocks (compared to in the no attention condition) while successfully remembering the rule to only respond to the cued stimulus set. Attentional tasks are normally designed to show some other measure of attentional modulation, for example by showing differing degrees of distraction in different situations. What basis in the literature is there for this definition of feature-based attention? Why is the tACS effect taken as showing "control of feature-based attention"? The key statistical result is the interaction between stimulation phase and sensory evidence, indicating "that the corresponding sensory information influences participant's behaviour more strongly in the in-phase stimulation condition compared to out of phase". This is then saying that visual discrimination performance is overall better with in phase versus out of phase – but why is this overall improvement interpreted as affecting "feature based attention".

To what extent is the IFJ always the most active area? It is claimed that in fMRI Experiment 1 that IFJ is the "most active brain area" – to support this claim all whole-brain activations should be shown e.g. in a table, rather than just one slice. On what measure is this the "most active": cluster size/ peak z-score/ etc? Similarly in the EEG analysis (Exp 2) it is reported that "the most prominent significant cluster was located at the level of the left IFJ". Again what were all results and on what basis was the IFJ most prominent? Furthermore, Left IFJ is tagged to stimulus visibility after beamforming (Fig 1 h) – were other areas also present in this contrast?

How much can the methods here justify spatial resolution high enough to specifically implicate the IFJ? Throughout, the spatial resolution of EEG – even with 128 channels - does not allow the spatial resolution of the conclusions made regarding specificity to IFJ (as opposed to other nearby cortical regions). EEG data even with beamforming cannot show that a response is in the IFJ. EEG just has too low spatial resolution to test whether "IFJ gets tagged to the rhythmic stimulus presentation". The spatial resolution of tACS is also too coarse to allow strong inferences about the IFJ, even with the concentric ring electrode arrangement.

No control site is used for tACS in Experiment 3, meaning that effects could be driven by the tactile sensation of the tACS being in phase or out of phase with the visual stimulation. EMLA cream is applied to reduce the sensation but no measurement is taken of how well this worked nor what participants felt. A control site is necessary for Experiment 3.

The control experiment Experiment 5 finds no effect of stimulating a different region but the active sites use 50% more participants (e.g. 33 compared to 22 if I have calculated the rejected participants correctly) – the control experiment needs to have the same statistical power for a null to be more meaningful.

The eye tracking analysis focuses only on eye movements smaller than one degree, and even more specifically on their frequency. What are the results of an identical analysis including larger eye movements, and particularly their amplitude, for all experiments? The main task showed participants moving dots or scenes without fixation and it is likely that participants made eye movements differently to these, and in comparison to the fixation control task.

MINOR COMMENTS

The “no attention task” was always performed first – does this mean that differences between the two blocks could be attributed to order effects?

The “no attention task” is a central monitoring task whereas the attention tasks require discriminating larger-visual-field stimuli. Is one likely interpretation of the difference between the two tasks then rather the spatial distribution of attention (peripheral versus central)?

In the task participants are told to only respond to the last stimulus in the sequence of 4-7 stimuli. This suggests that a likely tactic is that they do not use the attentional cues for at least the first 3 trials. Why should it be inferred that participants also attend throughout?

The ERP data for Experiment 2 shown in 1g does not correspond to what is described in the legend: the figure shows “source- IFJ” on the left and “Frontal electrodes” on the right whereas the legend gives “scalp sensor (top) and source (bottom) and goes on to describe some occipital data which is not present. What is the number of bad channels and noisy trials removed based on visual inspection (Pg8).

The voxel with the highest dWPLI near the IFJ and near the visual cortex are chosen –what is the definition of ‘near’ here (Pg 8).

In Experiment 2 it is claimed that “most of the visual cortex gets entrained” on the basis of EEG data – however EEG data in general, and the dWPLI values in specific, cannot be taken to show how much visual cortex is being entrained.

Reviewer #3:

Remarks to the Author:

In the present manuscript, the authors employ a multimodal approach to infer if activity fluctuations in the inferior frontal cortex are related to attentional fluctuations over time. To establish a link, the authors first use fMRI to show that IFJ is selectively activated, then use EEG to make inferences about the precise timing and lastly, in three experiments, employ non-invasive, frequency-matched brain stimulation to provide causal evidence. The three tACS studies are either based on in-/anti-phase stimulation, a more fine-grained binned (6-bins) stimulation approach and a control experiment (vertex stimulation). Moreover, the authors employ a range of behavioral and statistical modeling tools. Overall, the manuscript is well written, interesting for a broad neuroscience audience and sets a high bar for tACS experiments in general.

Given that the results are presented very concisely in a dense manuscript, several queries remain that need to be addressed prior to publication.

Abstract

- ‘Regulated by the ongoing phase of slow excitability fluctuations’ – this question cannot be addressed in the context where slow fluctuations are exogenously driven. Based on the EEG activity alone (exp. 2) this is not well supported. It’d be encouraging to see a spectrogram of connectivity between both IFJs during attention (without sensory confounds) that highlights a peak at 1.4 Hz (or in the delta-range)

Introduction

- How do the authors envision the link between the fluctuations that they study and e.g. rhythmic sampling of feature-based attention as described by Re & Landau (CurrBiol) in the context of rhythmic behavior (framework by Fiebelkorn & Kastner).

Results

- It remains unaddressed if the described EEG effects are frequency-specific and if fMRI and EEG actually converge on the same voxels. An overlay of combination of effect sizes across both modalities would be great to highlight the overlapping ROI.
- More detailed queries are outlined below in the context of the respective figures

Fig 1

- Panel b: contrast motion>scene or vice versa: is there any IFJ activation? From the chosen presentation, it is impossible to assess if there is an activation in the IFJ
- Panel c: How do the authors interpret the difference in ceiling performance (performance in last bin and

- slope of the evidence accumulation curve, which is steeper for scenes) between motion and scenes?
- Panel e: is this coherence (wpli) between the EEG signal filtered at 1.4 Hz (plus/minus a given range or is this broadband?). How would a coherence spectrogram of stimulus train vs. EEG across different frequencies look like?
 - Panel f: what is the corresponding source projection of the sensor level topography?
 - Panel g: where does the phase lag of the yellow trace (left, attention) relative to the dashed line (stimulus) come from? This lag is not evident for scalp data (right panel)
 - Panel h: how come that typical attention-related areas in the frontoparietal network (FEF, IPS, TPJ) do not show up in this contrast?
 - Panel i: is this the wpli to the stimulus train? I think the statistical contrast here might constitute double dipping, since this is centered on voxels that were already part of the cluster. Or are the chosen contrasts independent?

Fig 2

- Is there evidence from exp. 2 that IFJ needs to be bilaterally coupled to mediate behavioral benefits? I.e. can the authors study behavior in Exp. 2 as a function of connectivity (wpli) between the two IFJ ROIs? This would make the approach in Fig. 2 more credible
- Panel c: a contrast (between groups) to data reported in Fig. 1 would be desirable to see if the e.g., the slope of the evidence curve is steeper (at least, that's the impression)
- Panel g/h: could this analysis be contrasted against the naturally-occurring fluctuations as obtained e.g. from experiment 2? This would help to assess variability in performance over time.
- Panel h: it seems there is no benefit from in-phase stimulation, but rather a disruption due to anti-phase stimulation? Again, knowing the role of phase relationships between left and right IFJ might help to reconcile these findings. This could be performed on the EEG data from exp. 2

Fig. 3

- Stimulation over Cz and the behavioral results are very encouraging. It would be necessary to see that task performance (basic parameters as outlined before) are comparable and not present as a result of differences between groups
- A illustration of preferred phase delay per participants would be desirable to understand if this phase-dependence is the result of a lag between stimulation and ongoing activity or the result of relative synchrony between the two IFJs
- It would be great to assess the impact of phase-dependent stimulation vs. phase-dependence of the physiologically-occurring behavior. i.e. if a 90 deg. offset provides the best fit, then is that indicative how IFJ is best synchronized or is this effect really only due to the tACS waveforms?

Fig S2

- Quantification of microsaccades and representative examples could be shown

Fig S3

- Unclear how the waveform was shifted. The authors only state 'iming the tACS waveform a few ms before the tagged slow rhythmic fluctuation'

In sum, this manuscript employs several state-of-the-art methods to provide a multi-modal perspective into the role of IFJ for feature-based attention. Several queries remain, but in principle, this manuscript provides a valuable resource for future studies how to approach the functional role of EEG activity for behavior.

Author Rebuttal to Initial comments

Reviewer #1

Main Point

The authors present a new approach that combines a behavioral task, fMRI, EEG and non-invasive tACS brain stimulation to predict and experimentally control the excitability states of the inferior frontal junction (IFJ) during a non-spatial, 'feature-based' attention task.

The show that applying electrical stimulation (tACS) to IFJ led to changes in performance, indicating a causal role of the phase-alignment in this region of prefrontal cortex. This effect was

further replicated in another experiment that did not require steady-state tagging of the IFJ through visual stimulation, but allowed for the study of systematic phase delays between tACS and the visual stimulus to be processed.

While the results for visual scenes were not significant in the original experiment with fixed in- and out-of-phase stimulations, the subsequent experiments showed significant modulation of attention to scenes when using a broader variety of stimulation timings, possibly reflecting the more complex pattern of sensory features in scenes.

The manuscript is well-written and carefully prepared. In the sum of all its parts, it draws a detailed picture of the causal involvement of IFJ in feature-based top-down attention. The authors invested a lot of effort in optimizing their stimulation paradigms and combining different imaging and electrophysiological measures within this one study. The conclusions that are drawn from the experiments are all well-supported by the data and go far beyond our current knowledge of this special region in PFC.

Resp: We thank the reviewer for the positive assessment of our work and for bringing awareness to various aspects that we carefully considered in our revision

- 1) Despite all these positive aspects, I think there is some general confusion about feature- vs. object-based attention. This confusion is already present in the Introduction of the paper, but continues also in the Methods, and in the interpretation of the results. Given the rather complex task the authors chose to study IFJ's involvement in top-down attention, I think it would help to set a better theoretical foundation of the various aspects of attentional control in which IFJ might be involved. Particularly, the task combines a traditional feature-based attention task (in form of the coherent motion stimuli) with an object-based attention task (in form of the scene classification task). The key difference is that scenes are complex visual stimuli that are made up of many different features, often even of different feature dimensions. Although these are rather different tasks from an attentional point of view, the combination is justified in the sense that a) IFJ is involved in both feature- and object-based attention, and b) they are both non-spatial forms of attention, which is in stark contrast to classical spatial attention paradigms, in which IFJ is less involved. Therefore, the emphasis of the study seems to be more on the specific aspect of studying non-spatial attention – in contrast to spatial attention paradigms that are supposed to be governed by a dorsal attention system implying FEF and PPC instead. Actually, the authors might find it helpful to have a look into a recent review by Bedini & Baldauf, 2021 EJN, who described in much detail the division of labor between IFJ and FEF (both within PFC) in governing non-spatial (i.e. feature- and object-based) and spatial attention, respectively.*

Resp: We want to thank the reviewer for their insight. We agree that we have been imprecise in the definitions of the different forms of attention. We have followed the reviewers' suggestion and moved away from feature-based attention and replaced it with non-spatial attention throughout the manuscript (including the title of our work). Importantly, we have added a sentence to the introduction indicating the division between feature- vs. object- vs. non-spatial attention, and additionally we now refer to the recent review by Bedini & Baldauf to emphasize the anatomical differences with spatial attention. The corresponding revised section now reads (page 1, left column, 2nd paragraph)

Non-spatial attention is commonly subdivided into feature-based attention, focusing on one single feature (such as a direction of motion, color, or orientation), and object-based attention in which a participant attends a combination of features (such as an object or a scenery). There is consensus that this process is not an intrinsic property of sensory areas but relies on long-range functional interactions with prefrontal structures. A large body of work implicates the inferior frontal junction (IFJ) as a key source of control signals for both forms of top-down non-spatial attention [2-4], in contrast, spatial attention has been shown to be governed by a dorsal attention system involving the frontal eye fields (FEF) and posterior parietal cortex (PPC) [4]. However, the causal mechanisms of top-down regulation of non-spatial attentional control remain unclear.

2) Also in the Introduction, the authors motivate their study by referring to previous research that has shown that attentional performance fluctuates over time and that control structures might provide their feedback signals to visual cortex rhythmically. This is true, however, I think all the experimental evidence for this notion stem from paradigms that used spatial attention tasks. Therefore, the current study might be the first one actually to show that such rhythmic attentional sampling occurs also for non-spatial, feature-based attention.

Resp: We have been hesitant to claim that we are the first to show rhythmic attentional sampling for non-spatial attention, since we believe that two of the papers we cite (citations [4,5] in the revised submission) do actually employ object-based attention. However, we agree they still feature some spatial component in their paradigm, therefore we now write in the manuscript that to our knowledge we are the first to focus on rhythmic control in non-spatial attention: (Page 1, right column, 1st paragraph).

'Additionally, the role of rhythmic control with a focus on non-spatial attention (i.e., without spatial confounds) has not been previously established'

3) The Discussion section of the manuscript is rather short and ends somewhat abruptly, leaving some questions and implications of the study open. For example, what are the underlying anatomical and functional connectivities of IFJ, given its important (and here even causal) role in the mentioned attention tasks? In the end, the function of a brain area is determined by its intrinsic and extrinsic connectivity fingerprints. Some first indications for tight anatomical connections between IFJ and feature-/object-processing visual areas in IT cortex (as assessed with diffusion imaging for example) have already been provided by Baldauf & Desimone, 2014, showing that the more ventral parts of lateral PFC have tight connections with scene-processing areas like PPA. Further, Bedini and colleagues (Bedini et al., 2021 Cognitive Processing), presented recently some novel data on IFJ's anatomical connectome (based on probabilistic diffusion imaging tracking), showing its preferential connectivity with the ventral visual pathway (which is the one crucial for feature-based attention), particular in contrast to spatially organized control sites like FEF. They concluded from their tractography data that also in PFC there is a subdivision in a dorsal and ventral part responsible for spatial vs non-spatial top-down signals, respectively. On the functional connectivity side, IFJ has been shown to have systematic fMRI co-activations (as assessed with the help of meta-analytic connectivity modeling, MACM) with the ventral visual pathway containing areas that process high-level feature and object information (Bedini et al., biorxiv/Brain, Structure and Function, 2023). Furthermore, an analysis of IFJ's functional connectivity profile in resting state MEG has shown that it is tightly connected with many important high-level visual areas processing feature information, especially in the delta, beta, and gamma frequencies (see Soyuhos & Baldauf, EJM 2023).

Resp: We agree with the reviewer that the article would benefit from a paragraph on the connectivity of the IFJ. We have now included the following paragraph in the discussion: (Page 7, left column, 2nd paragraph).

'Our results show that the IFJ is causally involved in top-down non-spatial attention, which is in line with what is known about the anatomical and functional connectivity of the IFJ. It has been shown that the IFJ has systematic fMRI co-activations with the ventral visual pathway, areas that are involved in high-level non-spatial attention [10]. Based on maps of probabilistic connectivities [24] it has been shown that the IFJ has a high connection probability with both the fusiform face area (FFA) and the PPA [2]. Interestingly, the coherence between these areas increases in a tagging frequency as well as in high-gamma frequencies when attention to houses (PPA) and attention to faces (FFA) is exerted [2]. Furthermore, studies of spontaneous activity as measured with magnetoencephalography recordings show that the IFJ has a strong power coupling with the ventral visual stream in delta, beta, and gamma oscillations [25]. It is likely that we target the coherence of these types of top-down communication channels with our tACS paradigm via the modulation amplitude coupling fluctuations. It will be exciting to see future research pinpointing whether our

tACS paradigm exerts its influence through modulation of the coherence of the tagging frequency or other frequencies, like high-gamma using imaging methods more sensitive to high-frequency oscillations such as MEG. These findings could then be compared to studies of spatial attention, since the IFJ is often contrasted to the FEF, which connects more strongly to the dorsal visual stream and is thought to have a similar attentional control function in spatial attention [4,10,25].'

- 4) *Also in the discussion section, I was missing a paragraph on the functional specificity of IFJ's involvement in top-down attentional guidance. In particular, it has been proposed that many prefrontal regions, including IFJ, might belong to a more general 'multiple-demand system' (Duncan, 2010; Fedorenko et al., 2012). Here the reported activations seem much more focal and specific to IFJ itself. I think, a very important aspect that might explain the specific activation of IFJ in the current tasks is the carefully designed functional contrast in Fig. 1b, which used (as in Baldauf & Desimone, 2014) difficulty matched contrasts in fMRI. The matching of task difficulty might have helped in avoiding more wide-spread, and task un-specific activations in the contrasts 'Motion/Scene>no attention'.*

Resp: We agree with the reviewer that the reason for us to find such specific activations in the IFJ is likely due to the carefully designed contrast of attention vs. no-attention. We agree that the IFJ's involvement in the multiple demand system is also of interest and therefore we have included the following paragraph in the discussion: (Page 7, right column, 2nd paragraph)

'It is important to highlight that the IFJ might be involved in more processes than just non-spatial attention. It is proposed that many prefrontal regions work together in a general multiple-demand system that is particularly important when solving novel tasks that require, for instance, fluid intelligence [27]. What role the IFJ plays within this multiple demand system and if this is also related to non-spatial attention remains a question for future research; a question that could be tackled based on similar causal manipulations as the ones employed in this study.'

- 5) *I couldn't find anywhere the exact coordinates of the peak activations of IFJ. It would be helpful for future studies to provide the x/y/z locations. Do they fit well with the accurate location of IFJ, as for example, recently identified by Bedini and colleagues using activation likelihood estimations (ALE) across a large set of fMRI localizer studies (Bedini et al., biorxiv/Brain, Structure and Function, 2023)?*

Resp: For the right IFJ our peak activation occurs at 54, 10, 36 for the left at -42, 2, 30 (MNI space). This fits well with the fMRI localizer study of Bedini et al. [6]: right IFJ 46, 12, 28 left IFJ -42, 6, 30. We added the following sentence to the results: Page 2, right, 2nd paragraph.

'Peak activations in the contrast of attention > no-attention occurred at MNI = 54, 10, 36 and MNI = -42, 2, 30, which fit well with the location of the IFJ in the literature [10] (Supplementary Fig 2, Supplementary Tables 1-5).'

Furthermore, we have added a supplementary figure showing the coordinates of the IFJ as found by us and by Bedini as well as tables where we list significant clusters resulting from our whole brain analyses (Pages 16 and 26 in the revised manuscript).

Figure R1. The coordinates of the IFJ found in our fMRI experiment match the location of the IFJ in the literature well. a We find a peak activation (highest Z-score) at MNI = 54, 10, 36, for the right IFJ. b For the left IFJ we find a peak activation at MNI = -42, 2, 30. c Bedini et al [6] used activation likelihood estimations across a large set of fMRI localizer studies to find the right IFJ at MNI = 46, 12, 28. d The left IFJ was found at MNI = -42, 6, 30.

Below we show the table of the main fMRI clusters (clusters larger than 100 voxels) for Attention > No-Attention. Z-statistic images were thresholded at $Z > 4.0$ and a cluster correction was applied at a threshold of $P < 0.05$. The clusters are listed in order of largest cluster size, the location is given in MNI coordinates and the naming is taken from the automated anatomical labelling atlas (AAL3v1).

Table 1. Attention > No-attention

Cluster	Name	LH/RH	Location	Z-value	Voxels
1	Occipital Mid	R	38, -82, 6	7.0	11675
2	Parietal Sup	L	-22, -72, 56	5.7	558
3	Supp Motor Area	-	0, 4, 52	5.5	408
4	Parietal Sup	R	26, -66, 56	6.4	390
5	Precentral (IFJ)	R	54, 10, 36	5.8	387
6	Precentral	L	-38, -6, 62	4.9	343
7	Insula	R	34, 22, 2	5.7	269
8	Insula	L	-36, 24, 2	5.5	215
9	Precentral (IFJ)	L	-42, 2, 30	5.8	196
10	Cerebellum Crus2	-	2, -82, -32	4.8	170
11	Putamen	L	-22, 4, 8	4.7	121

Reviewer #2:

SUMMARY

The aim of this study is to show that exogenously modulating prefrontal excitability leads to phase-specific modulations in feature-based attention. Previous work linking prefrontal excitability and oscillations to feature-based attention have largely been correlative. This manuscript comprises five separate studies using either fMRI, EEG or tACS, and in addition modelling. In the main task throughout, participants see sequences of visual stimuli in which moving dots are superimposed onto pictures of scenes. A cue starting each trial instructs whether to discriminate dot direction or scene category. Then participants respond to the last presentation in each sequence. In a separate 'no attention' control task, always performed first, participants respond to fixation cross changes. In Experiments 1- 3 the visual stimuli fluctuate in visibility at 1.4 Hz to allow measuring for entrainment.

Experiment 1 finds as would be expected that motion or scene-specialised cortical areas are activated most when reporting to that type of information, and a general attentional network including the IFJ is important for both, compared to the control task. Experiment 2 records 128-channel EEG during this task. Many electrodes show a signal entrained to the frequency of the visibility fluctuation (using a measure of phase synchronisation, dWPLI). A source analysis reveals a cluster in the EEG data estimated to be around the IFJ to correlate with the visibility fluctuation more during attention than the no-attention blocks. The phase lag of this frontal cluster was delayed compared to those from occipital electrodes. An eye tracking analysis shows that microsaccades cannot explain differences in brain activity between different conditions. Experiment 3 then applies frontal tACS with one electrode over the IFJ and a concentric ring of return electrode around it. In different sessions, participants receive tACS either in phase or out of phase with the visual stimulus fluctuations. In phase stimulation was only better than out of phase stimulation for the motion stimuli. The tACS effect is found to disappear after tACS ends is found in a drift diffusion model to specifically affect the rate of sensory evidence accumulation. Experiment 4 drops the visual fluctuations and explores the effect of 6 different tACS phases using either just motion stimuli or just scenes, finding phase effects on both. Experiment 5 repeats experiment 4 by stimulating a vertex control site but does not find behavioural effects.

MAJOR COMMENTS

- 1) This is in many ways an impressive manuscript using several techniques, but raises several queries which detract from the conclusions.

More support is needed to explain why these effects are taken as showing specifically feature-based attention. It seems that the lack of distraction by the other task (motion versus scene) is taken to indicate that participants are orienting attention to the cued feature. But participants could be enhancing the features of both stimulus sets during attention blocks (compared to in the no attention condition) while successfully remembering the rule to only respond to the cued stimulus set. Attentional tasks are normally designed to show some other measure of attentional modulation, for example by showing differing degrees of distraction in different situations. What basis in the literature is there for this definition of feature-based attention? Why is the tACS effect taken as showing "control of feature-based attention"? The key statistical result is the interaction between stimulation phase and sensory evidence, indicating "that the corresponding sensory information influences participant's behaviour more strongly in the in-phase stimulation condition compared to out of phase". This is then saying that visual discrimination performance is overall better with in phase versus out of phase – but why is this overall improvement interpreted as affecting "feature-based attention".

Resp: We thank the reviewer for his observation which we agree is important to clarify. We believe that the involvement of the IFJ in the top-down attentional processes is not solely demonstrated by any individual result, but rather emerges as a comprehensive picture from the combination of our carefully designed experiments and the existing literature on the IFJ.

It is important to note that there is a substantial body of literature highlighting the IFJ's role in non-spatial attention where both theory and psychophysical experiments highlight that sensory gain is enhanced in the relevant sensory structure [1-3,10]. Furthermore, our task was designed to be challenging (average performance ~70-75%) rendering it unlikely that they tracked both visual streams simultaneously. Consequently, attentional selection becomes crucial for optimal performance.

Our findings, utilizing fMRI and EEG, provide compelling evidence to support the involvement of the IFJ in attentional control. Specifically, we observed heightened IFJ activity during attentive states compared to inattentive states using fMRI, while EEG measurements demonstrated greater entrainment of the IFJ to visual frequency during attentive states. The convergence of these results leads us to conclude that modulating the IFJ through tACS stimulation directly influences performance via attentional control.

In our manuscript we acknowledged that while our brain stimulation protocol appears to induce robust attentional influences in motion discrimination performance, these results do not clarify whether these behavioral modulations are indeed specific to boosting the perception of sensory evidence. Therefore, we employed the drift-diffusion model (DDM), a well-established mathematical model of human choices that in our application allows the possibility of disentangling how the manipulation of IFJ excitability states affects latent variables corresponding to distinct components of the decision process, while allowing latent variables of the DDM to be parametrically modulated as a function of the stimulation condition. If the hypothesis holds that IFJ excitability modulations specifically affect the degree of efficiency at which sensory areas accumulate sensory evidence, then we would expect opposing stimulation protocols to selectively affect the rate of sensory evidence accumulation. We found that this was indeed the case.

Additionally, please note that we also investigated whether some of the above-mentioned differences in the modulation of top-down attentional control were related to our non-invasive brain stimulation intervention inducing oculomotor modulations. Analyses of eye tracking data show that there is no significant difference between the number of (micro)saccades in the different brain stimulation conditions (Supplementary Fig 6). Together, we believe that our oculomotor and modeling analyses provide evidence that stimulation-induced attentional modulations are specifically related to boosting the degree of efficiency at which sensory areas accumulate sensory evidence.

We acknowledge the reviewer's suggestion to combine tACS with imaging techniques such as fMRI, which would have been fascinating. Unfortunately, due to the technical limitations imposed by our high stimulation amplitudes and their interference with the fMRI filters (and related safety issues raised by the ethics committee), we were unable to conduct these experiments.

In summary, by drawing from both our own experimental findings and the existing literature, we believe that the evidence presented supports the involvement of the IFJ in attentional processes.

2) *To what extent is the IFJ always the most active area? It is claimed that in fMRI Experiment 1 that IFJ is the “most active brain area” – to support this claim all whole-brain activations should be shown e.g. in a table, rather than just one slice. On what measure is this the “most active”: cluster size/ peak z-score/ etc?*

Resp: We agree that this statement should have been backed up with information of the whole-brain analyses. We have added supplementary tables reporting all major significant clusters whole brain corrected at the cluster level (cluster size > 100 voxels). We have changed the text to now read: (Page 2, right column, 2nd paragraph)

In Experiment 1, we found that the bilateral IFJ was the most active prefrontal brain area (both in terms of cluster size and peak z-score) in the attention task compared to the no-attention task for each sensory

modality (peak $Z_{\text{motion}}=5.9$, $Z_{\text{scene}}=6.1$, $P<0.001$, $P<0.05$ cluster corrected, Fig. 1b), with a high degree of overlap across the two sensory modalities (conjunction analysis $Z>2.6$, $P<0.05$ cluster corrected, Fig. 1b). Peak activations in the contrast of attention > no-attention occurred at MNI = 54, 10, 36 and MNI = -42, 2, 30, which fit well with the location of the IFJ in the literature [10] (Supplementary Fig. 2, Supplementary Tables 1-5).'

- 3) *Similarly in the EEG analysis (Exp 2) it is reported that “the most prominent significant cluster was located at the level of the left IFJ”. Again what were all results and on what basis was the IFJ most prominent? Furthermore, Left IFJ is tagged to stimulus visibility after beamforming (Fig 1 h) – were other areas also present in this contrast?*

Resp: We agree that the EEG analysis should not be restricted to the IFJ. We refined our EEG source analysis by limiting the source search space to voxels located inside the brain. We report these clusters in a supplementary table, similar to the fMRI results. We have adapted the results accordingly. We now report: (Page 3, left column, 2nd paragraph)

'We computed the the debiased weighted phase lag index (dWPLI) at 1.43Hz between the sensor data and the visual stimulation signal. This measure captures how much the EEG is tagged to the visual stimulation. First, we compared these values between the attention and no-attention tasks across electrodes, and found clusters where the dWPLI was higher in the attention task ($T_{\text{max}}=4.81$ and $T_{\text{max}}=4.51$ for the occipital and frontal clusters, respectively, $P<0.01$ whole- brain cluster corrected, Fig. 1e; for the time series of an example participant see Fig. 1f; for dWPLI in the frequency see in Supplementary Fig. 4). We then computed the dWPLI at the source level by carrying out a whole-brain analysis (Methods). Without contrasting attention vs no-attention states, we found that posterior brain areas get entrained to the frequency of the visual input, where the degree of entrainment is higher for visual areas (Fig. 1g). Contrasting attention and no-attention, we found a significant cluster located near the left IFJ (Fig. 1h,i). Despite the well-known lack of spatial precision resulting from EEG source analyses, we found a remarkable degree of overlap between the resulting significant EEG and fMRI clusters (Supplementary Fig. 5 and Supplementary Table 6). The lateralized prefrontal cluster is located in the vicinity of the IFJ, and given the low spatial resolution of EEG, it is likely that this cluster is related to the IFJ which is clearly activated following our fMRI analyses.'

- 4) *How much can the methods here justify spatial resolution high enough to specifically implicate the IFJ? Throughout, the spatial resolution of EEG – even with 128 channels - does not allow the spatial resolution of the conclusions made regarding specificity to IFJ (as opposed to other nearby cortical regions). EEG data even with beamforming cannot show that a response is in the IFJ. EEG just has too low spatial resolution to test whether “IFJ gets tagged to the rhythmic stimulus presentation”.*

Resp: We thank the reviewer for this observation which motivated us to investigate this further. Despite the well-known lack of spatial precision resulting from EEG source analyses, we found a remarkable degree of overlap (although left-lateralized) between the resulting significant EEG and fMRI clusters (see figure below). Thus, we are confident that the conclusions drawn from the EEG source analyses are reasonable.

We state this result in the main text (see revised paragraph in response #3 to this reviewer, above; Page 3, left column, 2nd paragraph in the revised manuscript), and the newly added Supplementary Figure 5 and Supplementary Table 6.

5) *The spatial resolution of tACS is also too coarse to allow strong inferences about the IFJ, even with the concentric ring electrode arrangement.*

Resp: While it is true that the activated volume by tACS even with ring electrodes is several cubic centimeters, we have shown via the electric field modeling that our tACS protocol is relatively focal to the IFJ (Supplementary Figure 7). Furthermore, our fMRI results indicate that this specific task strongly activates the IFJ compared to other brain areas close to the IFJ. Additionally, we verified this hypothesis with the conduction of Experiments 4 and 5, which confirmed the relative focality of our target area (note that Cz is also typically used in TMS studies as an active control site). Therefore, we believe it is likely that our stimulation paradigm modulates behaviour by modulating activation dynamics in the IFJ or at the very least in its close vicinity.

6) *No control site is used for tACS in Experiment 3, meaning that effects could be driven by the tactile sensation of the tACS being in phase or out of phase with the visual stimulation. EMLA cream is applied to reduce the sensation but no measurement is taken of how well this worked nor what participants felt. A control site is necessary for Experiment 3.*

Resp: We carefully considered the reviewer's proposal and discussed it with the editor handling the manuscript in this journal. We believe that the scientific gain is marginal and the conclusions that could be derived from this result do not justify the monetary and time investments required when conducting this experiment. We base this on the following reasons:

1. Experiment 4 is the experiment where we more generally show evidence for top-down attention modulation. In this experiment, we show that the stimulation-induced attentional modulations do not require sensory tagging of the IFJ to rhythmic sensory manipulations (as done in Experiment 3). We reasoned that the relatively weak effect for scenes in experiment 3 might have been due to the multidimensional and non-local nature of the scene stimuli. In experiment 4 we showed that we could indeed successfully modulate attention in both sensory modalities due to our strategy of using a multi-phasic approach.
2. Based on this we conducted a new experiment (Experiment 5) to test whether the effects of tACS on feature-based attention observed in Experiment 4 were (i) specific to the IFJ, (ii) were not due to our tailored design to induce a generalized oscillatory sensory tagging in the brain, (iii) were not due by transcutaneous stimulation of peripheral nerves, and (iv) are not related to potential marginal influences of the electric field potentially reaching sensory areas. We identified (based on our neuroimaging data experiments) and stimulated a different brain structure to the IFJ that was in principle not related to feature-based attention (Cz location of the 10-20 EEG coordinate system). All other experimental parameters were equal to those of experiment 4. In this new experiment 5, we found no significant modulations of behavioral performance as a function of the phase of the tACS-induced electric field for motion or for scenes.
3. Given that the more general proof of causality in our work stems from experiment 4 and its proper control site experiment 5, we believe that performing an additional control for experiment 3 is not necessary nor sufficient relative to the evidence provided in experiments 4,5. In addition to these purely scientific reasons, a significant amount of monetary and time resources would have to be invested. Conducting a control experiment for experiment 3 would entail the recruitment of about n=40 new participants who would have to come to the lab for two separate sessions (that is, about 80 sessions conducted at least three days apart). Given the technical setup of the closed-loop system, this is an experiment that cannot be parallelized and would take several months to complete, a task that will be impossible to complete by the PhD students who have been leading the project for about 5 years already (Jeroen, Joseph, and Valeriia). Additionally, substantial amendments would have to be submitted to the ethical commission, causing further delays.

7) *The control experiment Experiment 5 finds no effect of stimulating a different region but the active sites use 50% more participants (e.g. 33 compared to 22 if I have calculated the rejected participants correctly) – the control experiment needs to have the same statistical power for a null to be more meaningful.*

Resp: We completely agree with the reviewer's observation and we have increased the number of participants in experiment 5 to match the number of participants in experiment 4. The conclusions have remained the same, with the empirical values being bigger than 59% and 67% of the null distribution for motion and scenes respectively.

In addition, we have performed a Bayes factor analysis to test if the amplitude in a sinusoidal model is larger than zero relative to the empirical null (see Methods). For experiment 4 we found $BF_{10} > 100$ for motion and $BF_{10} > 100$ for scenes, which indicates "Extreme" evidence against the null model. For experiment 5 we found Bayes Factors of $BF_{01} = 2.9$ and $BF_{01} = 2.7$ in favor of the null, which indicates "anecdotal" evidence for the null model (thus in the opposite direction to the BF analyses in experiment 4). In the revised version of the manuscript, we now report these results in the text at page 6 left column, 1st paragraph and page 6, right column, 1st paragraph.

8) *The eye tracking analysis focuses only on eye movements smaller than one degree, and even more specifically on their frequency. What are the results of an identical analysis including larger eye movements, and particularly their amplitude, for all experiments? The main task showed participants moving dots or scenes without fixation and it is likely that participants made eye movements differently to these, and in comparison to the fixation control task.*

Resp: Based on the reviewer's comment, we have carried out a more detailed eye-tracking analysis. We have separated the analysis for saccades (> 1 degree of visual angle) and microsaccades (<1 degree). As it can be appreciated in the figure below (now added as Supplementary Figure 6 in the revised manuscript), there are no significant differences between motion, scene, no attention trials neither in in-phase versus out-of-phase stimulation trials with regards to their quantity (b), amplitudes (c), or direction (d,e).

Figure R2. (Micro)saccade analysis shows no significant differences between motion, scene or no-attention trials, neither between in-phase or out-of-phase trials. **a** Representative example of the gaze position during one trial. The example shows 3 saccades and several microsaccades. **b** Analysis of the eye tracking data shows no significant differences in the number of saccades and microsaccades per stimulus presentation in which motion or scenes were cued, or in unattentive states. There are also no significant differences in saccades per stimulus for in-phase or out-of-phase stimulated trials (All pairwise combinations of paired t-tests $P > 0.4$). **c** The density of amplitudes show that there are no major differences in amplitudes between the motion, scene, non attentive, in-phase or out-of-phase trials. **d** The spider plots show that the direction of the microsaccades for the different conditions are similar. **e** Same as **d**, but for saccades > 1 degree.

MINOR COMMENTS

- 9) *The “no attention task” was always performed first – does this mean that differences between the two blocks could be attributed to order effects?*
- 10) *The “no attention task” is a central monitoring task whereas the attention tasks require discriminating larger-visual-field stimuli. Is one likely interpretation of the difference between the two tasks then rather the spatial distribution of attention (peripheral versus central)?*

Resp: Here we respond to points 9 and 10 as they are closely related. We cannot rule out that the differences between the two blocks are due to order effects. However, please note that performing the no-attention task first (before explaining the attention task) was necessary to make sure that participants would not already start performing the attention task as a way of practicing and preparing during the no-attention task.

The reviewer is correct that it is technically possible that the IFJ activates for peripheral attention as opposed to central attention. This is an interesting possibility that will be interesting to pursue in future studies, for example by adapting the current no-attention version of the task by allowing the button press cue to appear at a random location on the screen. Please note that the differences between the attention and no-attention activations in the fMRI images are relatively small, with the logical exceptions of MT+ and PPA, which is in line with the literature on motion detection and scene recognition. Moreover, all our causal manipulations were within the scope of the attention task and did not involve potentially spurious comparisons to a no-attention task. Nevertheless, we agree with the reviewer that this is an important aspect to highlight which we now include in the discussion of the revised manuscript: (Page 6, right column, 3rd paragraph)

‘The IFJ attention maps in our imaging analyses were obtained based on the contrast of an attention task (focused on either motion or scenes) versus a no-attention task (in which participants see the same visual stimuli, but only report the incidental rotations of a fixation cross). Given that we performed the no-attention task first, we cannot rule out the possibility that our imaging results can be explained by order effects. However, please note that performing the non-attentional task first (before explaining the attention task) was necessary to make sure that participants would not already start performing the attention task as a way of practicing and preparing during the no-attention task. Additionally, we can not exclude the possibility that the results of our IFJ causal manipulations are not specific for central attention as opposed to peripheral attention. This is an interesting possibility that should be pursued in future studies, for example, by adapting the current no-attention version of the task by allowing the button press cue to appear at a random location on the screen.’

- 11) *In the task participants are told to only respond to the last stimulus in the sequence of 4-7 stimuli. This suggests that a likely tactic is that they do not use the attentional cues for at least the first 3 trials. Why should it be inferred that participants also attend throughout?*

Resp: Although it is true that participants could relax during the first three stimuli and only start paying attention from the 4th stimulus onward, we show in Figure 1 panel g the event-related potential for the first four periods of the visual stimulus and observe that the entrainment to the visual signal starts at the first stimulus in both visual and frontal areas (both in the EEG and source analyses). This suggests that participants pay attention to the visual stimuli from stimulus-train onset. We also want to note that every stimulus is only presented for 0.7 seconds, which gives the participants the experience of a very fast-paced game. This gives little incentive for participants to not pay attention to the first stimuli. Out of our own experience during the creation and piloting of the task, we found based on feedback that it is less effortful to pay attention throughout all stimuli than to count the first three stimuli and then selectively pay attention to the 4th stimulus and onward.

12) *The ERP data for Experiment 2 shown in 1g does not correspond to what is described in the legend: the figure shows “source- IFJ” on the left and “Frontal electrodes” on the right whereas the legend gives “scalp sensor (top) and source (bottom) and goes on to describe some occipital data which is not present.*

Resp: We thank the reviewer for pointing this out. We have corrected it.

13) *What is the number of bad channels and noisy trials removed based on visual inspection (Pg8).*

Resp: We have clarified this point in the manuscript: (Page 9, left column, 3rd paragraph)

‘We removed 179 bad trials (100 in the attention and 79 in the no-attention tasks, corresponding to 4% and 6% of the trials, respectively) 5 bad channels based on visual inspection.’

14) *The voxel with the highest dWPLI near the IFJ and near the visual cortex are chosen –what is the definition of ‘near’ here (Pg 8).*

Resp: We have clarified this point in the manuscript: (Page 9, right column, 2nd paragraph)

“We identified for each subject the voxels with the highest dWPLI in the attention task within a sphere of 4cm radius centered at the frontal and occipital voxels with the highest dWPLI across participants”

15) *In Experiment 2 it is claimed that “most of the visual cortex gets entrained” on the basis of EEG data – however EEG data in general, and the dWPLI values in specific, cannot be taken to show how much visual cortex is being entrained.*

Resp: We agree with this comment. We have removed “most” and now write: (Page 3, left column, 2nd paragraph.)

“... we found that posterior brain areas get entrained to the frequency of the visual input, where the degree of entrainment is higher for visual areas”

Reviewer #3:

Remarks to the Author:

In the present manuscript, the authors employ a multimodal approach to infer if activity fluctuations in the inferior frontal cortex are related to attentional fluctuations over time. To establish a link, the authors first use fMRI to show that IFJ is selectively activated, then use EEG to make inferences about the precise timing and lastly, in three experiments, employ non-invasive, frequency-matched brain stimulation to provide causal evidence. The three tACS studies are either based on in-/anti- phase stimulation, a more fine-grained binned (6-bins) stimulation approach and a control experiment (vertex stimulation). Moreover, the authors employ a range of behavioral and statistical modeling tools. Overall, the manuscript is well written, interesting for a broad neuroscience audience and sets a high bar for tACS experiments in general.

Given that the results are presented very concisely in a dense manuscript, several queries remain that need to be addressed prior to publication.

Resp: We thank the reviewer for the constructive observations, and for finding our manuscript interesting for a broad audience. We reply point by point to their comments below

1) Abstract

'Regulated by the ongoing phase of slow excitability fluctuations' – this question cannot be addressed in the context where slow fluctuations are exogenously driven. Based on the EEG activity alone (exp. 2) this is not well supported. It'd be encouraging to see a spectrogram of connectivity between both IFJs during attention (without sensory confounds) that highlights a peak at 1.4 Hz (or in the delta-range)

Resp: We acknowledge the limitation of claims on endogenous oscillations given that fluctuations are exogenously manipulated in our study. In the revised version of the manuscript (abstract, introduction, and discussion), we are now careful about differentiating between endogenous oscillations and our exogenous manipulations. Our intention was not to claim that there is specifically a 1.4Hz endogenous oscillation. Our study shows that if the activity of the IFJ fluctuates, then so will the ability to exert non-spatial attention. With the design of our task, it is not possible to disentangle potential endogenous oscillations around 1.4Hz with the exogenously manipulated fluctuations, since the visual stimulation is always presented in a rhythmic manner. This was intentional, as this design allowed us to more accurately control the phase manipulation given the frequency tagging properties of our brains.

We would like to use this opportunity to clarify (given that the same possible misunderstanding is reflected in other comments below) that we do not claim that non-spatial attention relies on the connectivity between IFJs, but is dependent on the fluctuations of excitability of the IFJ. Please note that we stimulated both IFJs simultaneously to improve the chances of inducing behavioral modulations (given the bilateral IFJ occurrence found in the fMRI experiments), not to study the effects of synchronization or de-synchronization between the IFJs. In all paradigms, both IFJs were stimulated identically, also with identical phase delay as compared to the visual input.

Therefore, we did not manipulate the connectivity between IFJs, but rather the responsiveness of both IFJs with the externally-induced sensory tagging. The idea of the reviewer may be an interesting idea to pursue in future research, that is, what would be the result of desynchronizing both IFJs?

In the revised version of the manuscript, we now rewrite the following paragraph in the discussion section to clarify the endogenous vs exogenous discussion: (Page 7, left column, 3rd paragraph)

'While in our work we do not study the role of endogenous fluctuations but control them exogenously, our

paradigm and results provide evidence that prefrontal excitability states are causally related to guide top-down attention. Here, we acknowledge that with our paradigm we cannot distinguish between modulating endogenous neural activity or modulation of phasic activity on top of the exogenously controlled oscillation. However, irrespective of this consideration, our results support the theory that non-spatial attention relies on ongoing prefrontal excitability states which are likely regulated by slow oscillatory dynamics that guide goal-oriented behavior. Following up on our predator example, our findings entail that if the direction of the predator's movement behind the bush is the relevant feature, high excitability states in prefrontal structures regulating top-down attention would promote correct discrimination of the predator's direction of movement.'

2) Introduction

- *How do the authors envision the link between the fluctuations that they study and e.g. rhythmic sampling of feature-based attention as described by Re & Landau (CurrBiol) in the context of rhythmic behavior (framework by Fiebelkorn & Kastner).*

Resp: The reviewer is right in that previous studies provide compelling indications for the role of slow oscillations in providing windows of opportunity for attention [4,5,12,13]. Shown is that both in spatial attention as well as in non-spatial attention locations and features are rhythmically sampled. This means that even for instructed (or self guided) sustained attention, there are periods of enhanced and diminished perceptual sensitivity that seem to be driven by slow oscillations. One possibility is that this helps higher-order brain areas to optimally use their limited neural resources, by segregating in time their functional connectivity with either sensory or motor areas. Periods in which functional connectivity with sensory areas is relatively high correspond to sampling windows and high perceptual sensitivity while periods with high connectivity to motor areas correspond to periods in which attention is switched to a different location, feature or object.

Our work strongly relates to this literature because we find that the IFJ rhythmically activates when driven by a periodic visual stimulus. Furthermore, when modulating this entrainment with tACS we find that it is possible to modulate attentional behavior as a function of the phase of the electrical stimulation. This underlines the rhythmicity of attention and the importance of synchronizing neuronal activity with visual input for optimal perceptual sensitivity. We now write in the discussion: (Page 7, left column, 3rd paragraph)

'Temporal manipulations of sensory evidence in recent behavioral and neuroimaging studies led researchers to hypothesize that slow periodic neuronal excitability fluctuations in prefrontal structures shape the temporal dynamics of attention [5-8]. These studies show that both in spatial attention as well as in non-spatial attention locations and features are rhythmically sampled. This means that even during instructed (or also perhaps intended) sustained attention there are periods of enhanced and diminished perceptual sensitivity. One possibility is that this helps higher-order brain areas to optimally use their limited neural resources, by segregating in time their functional connectivity with either sensory or motor areas. Periods in which functional connectivity with sensory areas is relatively high correspond to sampling windows and high perceptual sensitivity while periods with high connectivity to motor areas correspond to periods in which attention is switched to a different location, feature or object [26]...'

3) Results

- *It remains unaddressed if the described EEG effects are frequency-specific and if fMRI and EEG actually converge on the same voxels. An overlay of combination of effect sizes across both modalities would be great to highlight the overlapping ROI.*

Resp: Thank you for making this suggestion which we also believe is important to investigate (we address the question of the frequency specificity of the EEG results below, see response to comment 6). We have added a supplementary figure with the overlay of fMRI and EEG results (Supplementary Figure 5 in the revised manuscript). Due to the spatial imprecision of the EEG source analysis, the results don't exactly converge onto the same voxels, however, we believe it is reasonable to assume the activations do in fact originate from the left IFJ. Page 19.

Figure R3. Overlay of fMRI activations and EEG source analysis results. In a contrast of attentive versus non-attentive states we find that both the left and right IFJ activate in the fMRI experiment, in the EEG source analysis the results are lateralized towards the left IFJ. The EEG cluster is in the vicinity of the IFJ, but does not exactly overlap, likely due to the lower spatial resolution of EEG.

More detailed queries are outlined below in the context of the respective figures

4) Fig 1

- Panel b: contrast motion>scene or vice versa: is there any IFJ activation? From the chosen presentation, it is impossible to assess if there is an activation in the IFJ

Resp: Because the IFJ activates in a similar manner both for attention to motion and attention to scenes, this activation does not show up in a contrast of the two. We agree with the reviewer that this is valuable information the reader should have access to, therefore we have included 5 supplementary tables with significant clusters for the different contrasts. (pp. 26-27 in the revised manuscript).

5) Panel c: How do the authors interpret the difference in ceiling performance (performance in last bin and slope of the evidence accumulation curve, which is steeper for scenes) between motion and scenes?

Resp: We believe that this is purely the result of the design and difficulty of the experimental paradigm, and we believe that does not lead to any fundamental insights. Ideally, these curves are as similar as possible, but due to individual differences, this is hard to achieve. It turns out that there was a wider range of difficulty for the scene trials than for the motion trials, with the easiest scene trials being easier than the easiest motion trials. We used the results of our pilot experiments as well as from the imaging results to calibrate as much as possible the difficulty in the tACS experiments (which we managed to achieve to a good degree, we believe).

6) Panel e: is this coherence (wpli) between the EEG signal filtered at 1.4 Hz (plus/minus a given range or is this broadband?). How would a coherence spectrogram of stimulus train vs. EEG across different frequencies look like?

Resp: The dWPLI analysis is based on the filtered EEG signal at the frequency of interest (1.43Hz). Nevertheless, as suggested by the reviewer, we now visualize the results of the dWPLI analyses between the visual stimulation signal and the EEG signal across different frequencies for the electrodes of the frontal and occipital clusters. These plots show that the most prominent dWPLI values occur at the target frequency, which subsequent lower value peaks at the corresponding harmonics. More importantly, please note that the difference between attention and no-attention conditions is also more prominent at the target frequency. We have added this plot as a supplementary figure in the revised manuscript (Supplementary Figure 4, Page 18)

Figure R4. dWPLI spectrogram. dWPLI values for different frequencies between the sensor data and the visual stimulation signal for the frontal and occipital cluster of electrodes (see Fig. 1e in the main text) for the attention (blue) and no-attention (red) tasks. The peaks correspond to the frequency of the visual stimulation signal (1.43 Hz, represented by the dashed line) and its harmonics. Shaded areas represent one standard error of the mean.

7) *Panel f: what is the corresponding source projection of the sensor level topography?*

Resp: The sensor level topography shows the contrast of the dWPLI between the attention and no-attention task. The source projection of this contrast are shown in panel h. We have reorganized the panels of the figure and rewrote the figure legend to clarify this relation.

8) *Panel g: where does the phase lag of the yellow trace (left, attention) relative to the dashed line (stimulus) come from? This lag is not evident for scalp data (right panel)*

Resp: This is a good observation by the reviewer for which we did not find a definitive explanation. A possibility is that frontal scalp sensors tend to aggregate the bilateral signals toward central electrodes, whereas source data was taken from the dipole axis with the highest power (as it is common practice in EEG source analyses). But we acknowledge that there might be other likely explanations.

9) *Panel h: how come that typical attention-related areas in the frontoparietal network (FEF, IPS, TPJ) do not show up in this contrast?*

Resp: In general terms these areas are involved in visual attention but not in the specific type of visual attention that is needed in the current behavioral paradigm. Specifically, for both the FEF and the IPS it is known that they are involved in spatial attention and directing attention through saccadic eye movements [3,6,8]. Our findings are also supported by our saccade analyses which showed that saccadic activity was reduced in the attention condition relative to no attention (see response to comment 18 below). Regarding the TPJ, the literature indicates that this structure is mostly involved in bottom-up attention [16]. When attention is focused on a specific location or object, sudden changes or unexpected stimuli in the periphery can capture attention and cause a shift in focus. The TPJ plays a role in detecting these unexpected events and redirecting attention towards them. So these brain areas are expected to show activity in a different type of behavioral task than was performed by our participants.

10) *Panel i: is this the wpli to the stimulus train? I think the statistical contrast here might constitute double dipping, since this is centered on voxels that were already part of the cluster. Or are the chosen contrasts independent?*

Resp: The barplot in panel i does show the dWPLI difference in the voxel of the near IFJ cluster that is selected based on the attention to no-attention contrast. We agree with the reviewer that we should be careful in claiming statistical significance here. The goal of the statistical contrast was to illustrate the resulting wPLI of the region of interest discovered in our whole brain source analyses. In the revised figure we removed the statistical significance results in the corresponding panel and also revised the figure legend accordingly.

11) *Fig 2- Is there evidence from exp. 2 that IFJ needs to be bilaterally coupled to mediate behavioral benefits? I.e. can the authors study behavior in Exp. 2 as a function of connectivity (wpli) between the two IFJ ROIs? This would make the approach in Fig. 2 more credible*

Resp: As clarified in our response to comment 1), we do not believe that the synchrony between the IFJs is critical to carry out non-spatial attention. Rather, we claim that the activity of the IFJ is critical to carry out non-spatial attention. However, we do agree that it is a good idea to investigate

if higher dWPLI values are related to higher accuracies. Unfortunately, as dWPLI requires multiple trials to be computed, it is difficult to investigate the variability on a trial-by-trial level in a reliable manner. However, we can group the trials into correct and incorrect responses and compute the difference in dWPLI. Although the dWPLI was slightly higher in correct trials, this difference was not significant ($T = 1.28$, $p = 0.22$). It is unclear if this lack of significance is due to the absence of an effect or a lack of sample size. However, it would be fine for us to include this analysis in the manuscript if the reviewer and editor consider it essential.

12) - Panel c: a contrast (between groups) to data reported in Fig. 1 would be desirable to see if the e.g., the slope of the evidence curve is steeper (at least, that's the impression)

Resp: Indeed there are differences in the slope of the performance/evidence curve. In the figure below we show the standardized coefficients of a multifactor logistic regression of task performance as a function of evidence levels (see figure below). We believe that this is purely the result of the design and difficulty of the experimental paradigm which might be perceived different across experimental conditions (lighting conditions, scanner noise, etc). We acknowledged that, ideally, these curves are as similar as possible, but due to individual differences, this is hard to achieve. Nevertheless, we would like to note that our strategy when planning the tACS experiments was to use the results of our pilot experiments as well as from the imaging results to calibrate as much as possible the difficulty in the tACS experiments across the motion and scene trials, which we managed to achieve to a good degree as it can be appreciated from the similar std. slope effect values in the two tasks.

Figure R5. The participants correctly used the available visual evidence in all experiments

13) Panel g/h: could this analysis be contrasted against the naturally-occurring fluctuations as obtained e.g. from experiment 2? This would help to assess variability in performance over time.

Resp: We have added a supplementary figure to show the naturally-occurring fluctuations in performance over time in experiments 1 and 2. We plot the main effect of sensory evidence on choice in a linear mixed-effect model including both the cued and uncued sensory evidence values in the same model (Supplementary Figure 9 in the revised manuscript, page 23). The results show that performance is stable over time. Note the difference with Figure 2 g in which the interaction effect between sensory evidence and stimulation condition was plotted and we see a clear effect of the stimulation.

Figure R6. Performance is stable over time. We show naturally occurring fluctuations in performance in time as the main effect of sensory evidence on choice in a linear mixed effects model including both the cued and uncued sensory evidence values in the same model. Note the difference with Fig. 2g of the manuscript in which the interaction effect between sensory evidence and stimulation condition was plotted and we observe a clear effect of the stimulation. The grey shaded area indicates the windows for which stimulation was turned on. Shaded areas around the lines indicate ± 1 SD of the posterior estimate. a Experiment 1: fMRI. b Experiment 2: EEG.

14) - Panel h: it seems there is no benefit from in-phase stimulation, but rather a disruption due to anti-phase stimulation? Again, knowing the role of phase relationships between left and right IFJ might help to reconcile these findings. This could be performed on the EEG data from exp. 2

Resp: As we clarified in our response to comment 1), we do not believe that the synchrony between the IFJs is critical to the conclusions of our work as we did not factorially manipulate IFJ synchrony in our experimental tACS conditions. It is true that the results point towards a disruption with anti-phase stimulation rather than an enhancement with in-phase stimulation, but note that this is relative to the frequency tag of the visual stimulus and the reaction of the cortex to this manipulation.

15) Fig. 3

- Stimulation over Cz and the behavioral results are very encouraging. It would be necessary to see that task performance (basic parameters as outlined before) are comparable and not present as a result of differences between groups

Resp: We have added panels in figure 3 to show the task performance in the control experiment. Panel i shows the performance in experiment 4, and panel k the performance in control experiment

5. Please see the corresponding figure in the main text (figure 3) to visualize the full figure.

Figure R7. h Experiment 4. Graph of the sinusoidal function of performance vs stimulation delay, with the estimated population-level parameters represented as a line, the shaded area indicates \pm SD. The dots represent the individual data for each participant per stimulation delay condition after being aligned for individually estimated phase delays and intersects. The vertical green and red bars indicate the time windows of best and worst performance, respectively. i Psychometric curves of the highest performance phase delay (green) and worst performance phase delay (red). j and k. Similar to h and i, but for Experiment 5.

16) - A illustration of preferred phase delay per participants would be desirable to understand if this phase-dependence is the result of a lag between stimulation and ongoing activity or the result of relative synchrony between the two IFJs.

Resp: As mentioned above we are not manipulating the relative synchrony between the IFJs, however, it is a good suggestion to study the preferred phase delay per participant. We find that the electrical stimulation is timed just before the peak activation of the IFJ for optimal performance. We have added the corresponding figure in a new supplementary figure showing this result in the revised manuscript (Supplementary Figure 11, Page 25).

Figure R8. The preferred timing of the electrical stimulation is slightly before the peak activation of the IFJ.

Here we plot the preferred phase delay between the visual and electrical stimulation per participant in blue. This is estimated by determining the peak of the sinusoidal function of performance versus stimulation delay in Experiment 4. At 0 ° the visual stimulation starts (grey) in yellow the circular mean of the preferred stimulation delay is indicated at 7 ° for motion and 39 ° for scenes. The peak activation of the IFJ (150 ms later than the visual stimulation) is indicated in red. We see that, in line with experiment 3, the preferred timing of the stimulation is slightly before the peak activation of the IFJ.

17) - *It would be great to assess the impact of phase-dependent stimulation vs. phase-dependence of the physiologically-occurring behavior. i.e. if a 90 deg. offset provides the best fit, then is that indicative how IFJ is best synchronized or is this effect really only due to the tACS waveforms?*

Resp: As indicated above, it is outside the scope of the current study to investigate the phase dependence between the IFJs. As mentioned in our response to comment 1, we agree that this will be an interesting aspect to investigate in future studies.

18) *Fig S2*

- *Quantification of microsaccades and representative examples could be shown*

Resp: Based on the reviewer's suggestion, we have added a more comprehensive saccade analysis. In the new analyses, we separated the analysis for saccades (> 1 degree of visual angle) and microsaccades (<1 degree). Panel (a) shows a representative trial. We found that there are no quantitative and qualitative differences between motion, scene and no attention trials neither for in-phase versus out-of-phase trials with regards to their quantity (b), amplitudes (c), or direction (d&e). We have updated the corresponding figure in the revised manuscript (Supplementary Figure 6, page 20)

Figure R9. (Micro)saccade analysis shows no significant differences between motion, scene or no-attention trials, neither between in-phase or out-of-phase trials. a Representative example of the gaze position during one trial. The example shows 3 saccades and several microsaccades. b Analysis of the eye tracking data shows no significant differences in the number of saccades and microsaccades per stimulus presentation in which motion or scenes were cued, or in unattentive states. There are also no significant differences in saccades per stimulus for in-phase or out-of-phase stimulated trials (All pairwise combinations of paired t-tests $P > 0.4$). c The density of amplitudes show that there are no major differences in amplitudes between the motion, scene, non attentive, in-phase or out-of-phase trials. d The spider plots show that the direction of the microsaccades for the different conditions are similar. e Same as d, but for saccades > 1 degree.

19) Fig S3

- Unclear how the waveform was shifted. The authors only state 'iming the tACS waveform a few ms before the tagged slow rhythmic fluctuation'

Resp: Thank you for this suggestion. We modified the figure and figure legend accordingly where now we write: (Page 19)

'Based on the results obtained in our EEG experiment, we found that the visual cortex (VC) and the IFJ get entrained to the visual stimulation. The delay between visual stimulation and the response in the visual cortex was about 100 ms and 150 ms for the IFJ. Using photosensitive triggers on the monitor we could record the exact timing of the visual stimulation and compare it to the ongoing electrical stimulation. The timing of the trough of in-phase stimulation (mean = 95 ms after the trough of visual stimulation, SD = 9) is represented in green and out-of-phase (mean = 445 ms after the trough of visual stimulation, SD = 15) in red. By timing the tACS waveform roughly 50 ms before the tagged slow rhythmic fluctuations in visual and prefrontal areas, we hypothesized that we could maximize the influence of the stimulation on behavior.'

20) *In sum, this manuscript employs several state-of-the-art methods to provide a multi-modal perspective into the role of IFJ for feature-based attention. Several queries remain, but in principle, this manuscript provides a valuable resource for future studies how to approach the functional role of EEG activity for behavior.*

Resp: We thank the reviewer once again for all the constructive comments and suggestions, which resulted in revisions that we believe improve the comprehension of the methods, results, conclusions, and discussions derived from our work,

 ***** End of reviewers' comments *****

References cited in this letter

[1] Daniel Baldauf and Robert Desimone. Neural mechanisms of object-based attention. Science, 344(6182):424–7, apr 2014. ISSN 1095-9203. doi:10.1126/science.1247003

[2] Theodore P. Zanto, Michael T. Rubens, Arul Thangavel, and Adam Gazzaley. Causal role of the prefrontal cortex in top-down modulation of visual processing and working memory. Nature Neuroscience, 14(5):656–663, 2011. ISSN 10976256. doi: 10.1038/nn.2773

[3] Marco Bedini and Daniel Baldauf. Structure, function and connectivity fingerprints of the frontal eye field versus the inferior frontal junction: A comprehensive comparison. European Journal of Neuroscience, 54(4):5462–5506, 2021. ISSN 14609568. doi: 10.1111/ejn.15393.

[4] Ian C. Fiebelkorn, Yuri B. Saalmann, and Sabine Kastner. Rhythmic Sampling within and between Objects despite Sustained Attention at a Cued Location. Current Biology, 23(24): 2553– 2558, dec 2013. ISSN 0960-9822. doi: 10.1016/J.CUB.2013.10.063 .

[5] Randolph F. Helfrich, Ian C. Fiebelkorn, Sara M. Szczepanski, Jack J. Lin, Josef Parvizi, Robert T. Knight, and Sabine Kastner. Neural Mechanisms of Sustained Attention Are Rhythmic. Neuron, 99(4):854–865.e5, aug 2018. ISSN 08966273. doi: 10.1016/j.neuron.2018.07.032 .

[6] Marco Bedini, Emanuele Olivetti, Paolo Avesani, and Daniel Baldauf. Accurate localization and coactivation profiles of the frontal eye field and inferior frontal junction: an ALE and MACM fMRI meta-analysis. Brain Structure and Function, 228(3):997–1017, 2023. ISSN18632661. doi: 10.1007/s00429-023-02641-y .

[7] Zeynep M. Saygin, David E. Osher, Kami Koldewyn, Gretchen Reynolds, John D.E. Gabrieli, and Rebecca R. Saxe. Anatomical connectivity patterns predict face selectivity in

the fusiform gyrus. *Nature Neuroscience*, 15(2):321–327, 2012. ISSN 10976256. doi:10.1038/nn.3001 .

- [8] Orhan Soyuhos and Daniel Baldauf. Functional connectivity fingerprints of the frontal eye field and inferior frontal junction suggest spatial versus nonspatial processing in the prefrontal cortex. *European Journal of Neuroscience*, (June 2022):1114–1140, 2023. ISSN14609568. doi: 10.1111/ejn.15936.
- [9] John Duncan. The multiple-demand (MD) system of the primate brain: mental programs for intelligent behaviour. *Trends in Cognitive Sciences*, 14(4):172–179, 2010. ISSN 13646613. doi: 10.1016/j.tics.2010.01.004.
- [10] Zanto, T.P., Rubens, M.T., Bollinger, J. & Gazzaley, A. Top-down modulation of visual feature processing: the role of the inferior frontal junction. *Neuroimage* 53, 736–745 (2010).
- [11] Christian Keysers, Valeria Gazzola, and Eric-jan Wagenmakers. neuroscience to establish evidence of absence. *Nature Neuroscience*, 23(July), 2020. ISSN 1546-1726. doi: 10.1038/s41593-020-0660-4 .
- [12] Ayelet Nina Landau and Pascal Fries. Attention Samples Stimuli Rhythmically. *CurrentBiology*, 22(11):1000–1004, jun 2012. ISSN 0960-9822. doi:10.1016/J.CUB.2012.03.054.
- [13] Laura Dugué, Mariel Roberts, and Marisa Carrasco. Attention Reorients Periodically. *Current Biology*, 26(12):1595–1601, jun 2016. ISSN 0960-9822. doi: 10.1016/J.CUB.2016.04.046.
- [14] Ian C. Fiebelkorn and Sabine Kastner. A Rhythmic Theory of Attention. *Trends in Cognitive Sciences*, 23(2):87–101, 2019. ISSN 1879307X. doi:10.1016/j.tics.2018.11.009.
- [15] Geoffrey Brookshire. Putative rhythms in attentional switching can be explained by aperiodic temporal structure. *Nature Human Behaviour*, pages 1–12, jun 2022. ISSN 2397-3374. doi:10.1038/s41562-022-01364-0 .
- [16] Corbetta, M., Kincade, J., Ollinger, J. *et al.* Voluntary orienting is dissociated from target detection in human posterior parietal cortex. *Nat Neurosci* 3, 292–297 (2000). <https://doi.org/10.1038/73009>

Decision Letter, first revision:

26th April 2023

Dear Prof Polania,

Thank you once again for your manuscript, entitled "Causal phase-dependent control of feature-based attention in human prefrontal cortex," and for your patience during the peer review process.

Your manuscript has now been evaluated by 3 reviewers, whose comments are included at the end of this letter. Although the reviewers find your work to be of interest, they also raise some important concerns. We are interested in the possibility of publishing your study in *Nature Human Behaviour*, but would like to consider your response to these concerns in the form of a revised manuscript before we make a decision on publication.

To guide the scope of the revisions, the editors discuss the referee reports in detail within the team, including with the chief editor, with a view to (1) identifying key priorities that should be addressed in revision and (2) overruling referee requests that are deemed beyond the scope of the current study. We hope that you will find the prioritized set of referee points to be useful when revising your study. Please do not hesitate to get in touch if you would like to discuss these issues further.

First, we ask that you address two important concerns by Reviewer #2 regarding the lack of a control site in Exp 3 and the difference in sample size in Exp 5 (comments 4 and 5). Please [1] perform an additional control experiment that includes better control for the tactile stimulation induced by tACS and a proper control site. Please [2] use Bayes Factors to quantify support for the null hypothesis in Experiment 5. If the evidence in support of the null is inconclusive (see <https://www.ejwagenmakers.com/2015/BayesianAnalysisEnclopedia.pdf>), please increase the sample size to be the same as for Experiment 3 and use Bayesian statistics to quantify support for the null. (If you decide to continue using NHST for Experiment 5 after adding more participants, you will need to use sequential analyses to control for Type I error, which is inflated when repeated analyses are performed in NHST). We also ask that you [3] answer the questions regarding the specificity of the effects in IFJ given the spatial resolution of the methods used in the study (comment 3 by Reviewer #2 and comment 2 by Reviewer #3). Finally, please [4] clarify the definition of feature-based attention used in the manuscript (comment 1 by Reviewer #1, comment 1 by Reviewer #2).

In sum, we invite you to revise your manuscript taking into account all reviewer and editor comments. We are committed to providing a fair and constructive peer-review process. Do not hesitate to contact us if there are specific requests from the reviewers that you believe are technically impossible or unlikely to yield a meaningful outcome.

We hope to receive your revised manuscript within two months. I would be grateful if you could contact us as soon as possible if you foresee difficulties with meeting this target resubmission date.

- Include a "Response to the editors and reviewers" document detailing, point-by-point, how you addressed each editor and referee comment. If no action was taken to address a point, you must provide a compelling argument. When formatting this document, please respond to each reviewer comment individually, including the full text of the reviewer comment verbatim followed by your response to the individual point. This response will be used by the editors to evaluate your revision and sent back to the reviewers along with the revised manuscript.
- Highlight all changes made to your manuscript or provide us with a version that tracks changes.

[REDACTED]

We look forward to seeing the revised manuscript and thank you for the opportunity to review your work. Please do not hesitate to contact me if you have any questions or would like to discuss these revisions further.

Sincerely,

Giacomo Ariani
Editor
Nature Human Behaviour

Reviewer expertise:

Reviewer #1: Rhythmic feature-based attention, fMRI, EEG, TMS

Reviewer #2: Rhythmic feature-based attention, EEG, TMS

Reviewer #3: Rhythmic feature-based attention, EEG

REVIEWER COMMENTS:

Reviewer #1:

Remarks to the Author:

Review of Causal phase-dependent control of feature-based attention in human prefrontal cortex by Brus and colleagues.

The authors present a new approach that combines a behavioral task, fMRI, EEG and non-invasive tACS brain stimulation to predict and experimentally control the excitability states of the inferior frontal junction (IFJ) during a non-spatial, 'feature-based' attention task.

The show that applying electrical stimulation (tACS) to IFJ led to changes in performance, indicating a causal role of the phase-alignment in this region of prefrontal cortex. This effect was further replicated in another experiment that did not require steady-state tagging of the IFJ through visual stimulation, but allowed for the study of systematic phase delays between tACS and the visual stimulus to be processed.

While the results for visual scenes were not significant in the original experiment with fixed in- and out-of-phase stimulations, the subsequent experiments showed significant modulation of attention to scenes when using a broader variety of stimulation timings, possibly reflecting the more complex pattern of sensory features in scenes.

The manuscript is well-written and carefully prepared. In the sum of all its parts, it draws a detailed picture of the causal involvement of IFJ in feature-based top-down attention. The authors invested a lot of effort in optimizing their stimulation paradigms and combining different imaging and electrophysiological measures within this one study. The conclusions that are drawn from the experiments are all well-supported by the data and go far beyond our current knowledge of this special region in PFC.

Despite all these positive aspects, I think there is some general confusion about feature- vs. object-based attention. This confusion is already present in the Introduction of the paper, but continues also in the Methods, and in the interpretation of the results. Given the rather complex task the authors chose to study IFJ's involvement in top-down attention, I think it would help to set a better theoretical foundation of the various aspects of attentional control in which IFJ might be involved. Particularly, the task combines a traditional feature-based attention task (in form of the coherent motion stimuli) with an object-based attention task (in form of the scene classification task). The key difference is that scenes are complex visual stimuli that are made up of many different features, often even of different feature dimensions. Although these are rather

different tasks from an attentional point of view, the combination is justified in the sense that a) IFJ is involved in both feature- and object-based attention, and b) they are both non-spatial forms of attention, which is in stark contrast to classical spatial attention paradigms, in which IFJ is less involved. Therefore, the emphasis of the study seems to be more on the specific aspect of studying non-spatial attention – in contrast to spatial attention paradigms that are supposed to be governed by a dorsal attention system implying FEF and PPC instead. Actually, the authors might find it helpful to have a look into a recent review by Bedini & Baldauf, 2021 EJN, who described in much detail the division of labor between IFJ and FEF (both within PFC) in governing non-spatial (i.e. feature- and object-based) and spatial attention, respectively.

Also in the Introduction, the authors motivate their study by referring to previous research that has shown that attentional performance fluctuates over time and that control structures might provide their feedback signals to visual cortex rhythmically. This is true, however, I think all the experimental evidence for this notion stems from paradigms that used spatial attention tasks. Therefore, the current study might be the first one actually to show that such rhythmic attentional sampling occurs also for non-spatial, feature-based attention.

The Discussion section of the manuscript is rather short and ends somewhat abruptly, leaving some questions and implications of the study open. For example, what are the underlying anatomical and functional connectivities of IFJ, given its important (and here even causal) role in the mentioned attention tasks? In the end, the function of a brain area is determined by its intrinsic and extrinsic connectivity fingerprints. Some first indications for tight anatomical connections between IFJ and feature-/object-processing visual areas in IT cortex (as assessed with diffusion imaging for example) have already been provided by Baldauf & Desimone, 2014, showing that the more ventral parts of lateral PFC have tight connections with scene-processing areas like PPA. Further, Bedini and colleagues (Bedini et al., 2021 Cognitive Processing), presented recently some novel data on IFJ's anatomical connectome (based on probabilistic diffusion imaging tracking), showing its preferential connectivity with the ventral visual pathway (which is the one crucial for feature-based attention), particular in contrast to spatially organized control sites like FEF. They concluded from their tractography data that also in PFC there is a subdivision in a dorsal and ventral part responsible for spatial vs non-spatial top-down signals, respectively. On the functional connectivity side, IFJ has been shown to have systematic fMRI co-activations (as assessed with the help of meta-analytic connectivity modeling, MACM) with the ventral visual pathway containing areas that process high-level feature and object information (Bedini et al., *bioRxiv/Brain, Structure and Function*, 2023). Furthermore, an analysis of IFJ's functional connectivity profile in resting state MEG has shown that it is tightly connected with many important high-level visual areas processing feature information, especially in the delta, beta, and gamma frequencies (see Soyuhos & Baldauf, EJN 2023).

Also in the discussion section, I was missing a paragraph on the functional specificity of IFJ's involvement in top-down attentional guidance. In particular, it has been proposed that many prefrontal regions, including IFJ, might belong to a more general 'multiple-demand system' (Duncan, 2010; Fedorenko et al., 2012). Here the reported activations seem much more focal and specific to IFJ itself. I think, a very important aspect that might explain the specific activation of IFJ in the current tasks is the carefully designed functional contrast in Fig. 1b, which used (as in Baldauf & Desimone, 2014) difficulty matched contrasts in fMRI. The matching of task difficulty might have helped in avoiding more wide-spread, and task un-specific activations in the contrasts 'Motion/Scene>no attention'.

I couldn't find anywhere the exact coordinates of the peak activations of IFJ. It would be helpful for future studies to provide the x/y/z locations. Do they fit well with the accurate location of IFJ, as for example, recently identified by Bedini and colleagues using activation likelihood estimations (ALE) across a large set of fMRI localizer studies (Bedini et al., *bioRxiv/Brain, Structure and Function*, 2023)?

signed, Daniel Baldauf

Reviewer #2:

Remarks to the Author:

SUMMARY

The aim of this study is to show that exogenously modulating prefrontal excitability leads to phase-specific modulations in feature-based attention. Previous work linking prefrontal excitability and oscillations to feature-based attention have largely been correlative. This manuscript comprises five separate studies using either fMRI, EEG or tACS, and in addition modelling. In the main task throughout, participants see sequences of visual stimuli in which moving dots are superimposed onto pictures of scenes. A cue starting each trial instructs whether to discriminate dot direction or scene category. Then participants respond to the last presentation in each sequence. In a separate 'no attention' control task, always performed first, participants respond to fixation cross changes. In Experiments 1- 3 the visual stimuli fluctuate in visibility at 1.4 Hz to allow measuring for entrainment.

Experiment 1 finds as would be expected that motion or scene-specialised cortical areas are activated most when reporting to that type of information, and a general attentional network including the IFJ is important for both, compared to the control task.

Experiment 2 records 128-channel EEG during this task. Many electrodes show a signal entrained to the frequency of the visibility fluctuation (using a measure of phase synchronisation, dWPLI). A source analysis reveals a cluster in the EEG data estimated to be around the IFJ to correlate with the visibility fluctuation more during attention than the no-attention blocks. The phase lag of this frontal cluster was delayed compared to those from occipital electrodes. An eye tracking analysis shows that microsaccades cannot explain differences in brain activity between different conditions. Experiment 3 then applies frontal tACS with one electrode over the IFJ and a concentric ring of return electrode around it. In different sessions, participants receive tACS either in phase or out of phase with the visual stimulus fluctuations. In phase stimulation was only better than out of phase stimulation for the motion stimuli. The tACS effect is found to disappear after tACS ends is found in a drift diffusion model to specifically affect the rate of sensory evidence accumulation.

Experiment 4 drops the visual fluctuations and explores the effect of 6 different tACS phases using either just motion stimuli or just scenes, finding phase effects on both.

Experiment 5 repeats experiment 4 by stimulating a vertex control site but does not find behavioural effects.

MAJOR COMMENTS

This is in many ways an impressive manuscript using several techniques, but raises several queries which detract from the conclusions.

More support is needed to explain why these effects are taken as showing specifically feature-based attention. It seems that the lack of distraction by the other task (motion versus scene) is taken to indicate that participants are orienting attention to the cued feature. But participants could be enhancing the features of both stimulus sets during attention blocks (compared to in the no attention condition) while successfully remembering the rule to only respond to the cued stimulus set. Attentional tasks are normally designed to show some other measure of attentional modulation, for example by showing differing degrees of distraction in different situations. What basis in the literature is there for this definition of feature-based attention? Why is the tACS effect taken as showing "control of feature-based attention"? The key statistical result is the interaction between stimulation phase and sensory evidence, indicating "that the corresponding sensory information influences participant's behaviour more strongly in the in-phase stimulation condition compared to out of phase". This is then saying that visual discrimination performance is overall better with in phase versus out of phase – but why is this overall improvement interpreted as affecting "feature based attention".

To what extent is the IFJ always the most active area? It is claimed that in fMRI Experiment 1 that IFJ is the "most active brain area" – to support this claim all whole-brain activations should be shown e.g. in a table, rather than just one slice. On what measure is this the "most active": cluster size/ peak z-score/ etc? Similarly in the EEG analysis (Exp 2) it is reported that "the most prominent significant cluster was located at the level of the left IFJ". Again what were all results and on what basis was the IFJ most prominent? Furthermore, Left IFJ is tagged to stimulus visibility after beamforming (Fig 1 h) – were other areas also present in this contrast?

How much can the methods here justify spatial resolution high enough to specifically implicate the IFJ? Throughout, the spatial resolution of EEG – even with 128 channels - does not allow the spatial resolution of the conclusions made regarding specificity to IFJ (as opposed to other nearby

cortical regions). EEG data even with beamforming cannot show that a response is in the IFJ. EEG just has too low spatial resolution to test whether "IFJ gets tagged to the rhythmic stimulus presentation". The spatial resolution of tACS is also too coarse to allow strong inferences about the IFJ, even with the concentric ring electrode arrangement.

No control site is used for tACS in Experiment 3, meaning that effects could be driven by the tactile sensation of the tACS being in phase or out of phase with the visual stimulation. EMLA cream is applied to reduce the sensation but no measurement is taken of how well this worked nor what participants felt. A control site is necessary for Experiment 3.

The control experiment Experiment 5 finds no effect of stimulating a different region but the active sites use 50% more participants (e.g. 33 compared to 22 if I have calculated the rejected participants correctly) – the control experiment needs to have the same statistical power for a null to be more meaningful.

The eye tracking analysis focuses only on eye movements smaller than one degree, and even more specifically on their frequency. What are the results of an identical analysis including larger eye movements, and particularly their amplitude, for all experiments? The main task showed participants moving dots or scenes without fixation and it is likely that participants made eye movements differently to these, and in comparison to the fixation control task.

MINOR COMMENTS

The "no attention task" was always performed first – does this mean that differences between the two blocks could be attributed to order effects?

The "no attention task" is a central monitoring task whereas the attention tasks require discriminating larger-visual-field stimuli. Is one likely interpretation of the difference between the two tasks then rather the spatial distribution of attention (peripheral versus central)?

In the task participants are told to only respond to the last stimulus in the sequence of 4-7 stimuli. This suggests that a likely tactic is that they do not use the attentional cues for at least the first 3 trials. Why should it be inferred that participants also attend throughout?

The ERP data for Experiment 2 shown in 1g does not correspond to what is described in the legend: the figure shows "source- IFJ" on the left and "Frontal electrodes" on the right whereas the legend gives "scalp sensor (top) and source (bottom) and goes on to describe some occipital data which is not present.

What is the number of bad channels and noisy trials removed based on visual inspection (Pg8).

The voxel with the highest dWPLI near the IFJ and near the visual cortex are chosen –what is the definition of 'near' here (Pg 8).

In Experiment 2 it is claimed that "most of the visual cortex gets entrained" on the basis of EEG data – however EEG data in general, and the dWPLI values in specific, cannot be taken to show how much visual cortex is being entrained.

Reviewer #3:

Remarks to the Author:

In the present manuscript, the authors employ a multimodal approach to infer if activity fluctuations in the inferior frontal cortex are related to attentional fluctuations over time. To establish a link, the authors first use fMRI to show that IFJ is selectively activated, then use EEG to make inferences about the precise timing and lastly, in three experiments, employ non-invasive, frequency-matched brain stimulation to provide causal evidence. The three tACS studies are either based on in-/anti-phase stimulation, a more fine-grained binned (6-bins) stimulation approach and a control experiment (vertex stimulation). Moreover, the authors employ a range of behavioral and statistical modeling tools. Overall, the manuscript is well written, interesting for a broad neuroscience audience and sets a high bar for tACS experiments in general.

Given that the results are presented very concisely in a dense manuscript, several queries remain that need to be addressed prior to publication.

Abstract

- 'Regulated by the ongoing phase of slow excitability fluctuations' – this question cannot be addressed in the context where slow fluctuations are exogenously driven. Based on the EEG activity alone (exp. 2) this is not well supported. It'd be encouraging to see a spectrogram of connectivity between both IFJs during attention (without sensory confounds) that highlights a peak

at 1.4 Hz (or in the delta-range)

Introduction

- How do the authors envision the link between the fluctuations that they study and e.g. rhythmic sampling of feature-based attention as described by Re & Landau (CurrBiol) in the context of rhythmic behavior (framework by Fiebelkorn & Kastner).

Results

- It remains unaddressed if the described EEG effects are frequency-specific and if fMRI and EEG actually converge on the same voxels. An overlay of combination of effect sizes across both modalities would be great to highlight the overlapping ROI.
- More detailed queries are outlined below in the context of the respective figures

Fig 1

- Panel b: contrast motion>scene or vice versa: is there any IFJ activation? From the chosen presentation, it is impossible to assess if there is an activation in the IFJ
- Panel c: How do the authors interpret the difference in ceiling performance (performance in last bin and slope of the evidence accumulation curve, which is steeper for scenes) between motion and scenes?
- Panel e: is this coherence (wpli) between the EEG signal filtered at 1.4 Hz (plus/minus a given range or is this broadband?). How would a coherence spectrogram of stimulus train vs. EEG across different frequencies look like?
- Panel f: what is the corresponding source projection of the sensor level topography?
- Panel g: where does the phase lag of the yellow trace (left, attention) relative to the dashed line (stimulus) come from? This lag is not evident for scalp data (right panel)
- Panel h: how come that typical attention-related areas in the frontoparietal network (FEF, IPS, TPJ) do not show up in this contrast?
- Panel i: is this the wpli to the stimulus train? I think the statistical contrast here might constitute double dipping, since this is centered on voxels that were already part of the cluster. Or are the chosen contrasts independent?

Fig 2

- Is there evidence from exp. 2 that IFJ needs to be bilaterally coupled to mediate behavioral benefits? I.e. can the authors study behavior in Exp. 2 as a function of connectivity (wpli) between the two IFJ ROIs? This would make the approach in Fig. 2 more credible
- Panel c: a contrast (between groups) to data reported in Fig. 1 would be desirable to see if the e.g., the slope of the evidence curve is steeper (at least, that's the impression)
- Panel g/h: could this analysis be contrasted against the naturally-occurring fluctuations as obtained e.g. from experiment 2? This would help to assess variability in performance over time.
- Panel h: it seems there is no benefit from in-phase stimulation, but rather a disruption due to anti-phase stimulation? Again, knowing the role of phase relationships between left and right IFJ might help to reconcile these findings. This could be performed on the EEG data from exp. 2

Fig. 3

- Stimulation over Cz and the behavioral results are very encouraging. It would be necessary to see that task performance (basic parameters as outlined before) are comparable and not present as a result of differences between groups
- A illustration of preferred phase delay per participants would be desirable to understand if this phase-dependence is the result of a lag between stimulation and ongoing activity or the result of relative synchrony between the two IFJs
- It would be great to assess the impact of phase-dependent stimulation vs. phase-dependence of the physiologically-occurring behavior. i.e. if a 90 deg. offset provides the best fit, then is that indicative how IFJ is best synchronized or is this effect really only due to the tACS waveforms?

Fig S2

- Quantification of microsaccades and representative examples could be shown

Fig S3

- Unclear how the waveform was shifted. The authors only state 'iming the tACS waveform a few

ms before the tagged slow rhythmic fluctuation'

In sum, this manuscript employs several state-of-the-art methods to provide a multi-modal perspective into the role of IFJ for feature-based attention. Several queries remain, but in principle, this manuscript provides a valuable resource for future studies how to approach the functional role of EEG activity for behavior.

Author Rebuttal, first revision:

20th November 2023

Dear Dr Polania,

Thank you for submitting your revised manuscript "Causal phase-dependent control of non-spatial attention in human prefrontal cortex" (NATHUMBEHAV-23030816A). It has now been seen by the original referees and their comments are below.

As you can see, two reviewers found that the paper had improved in revision, but Reviewer #2 had serious remaining concerns. Therefore, we decided to consult with Reviewer #3 about these issues. While Reviewer #3 agreed that the points raised by Reviewer #2 are valid, they suggest that they can be addressed with careful re-wording and toning down some of the claims. We will therefore be happy in principle to publish it in Nature Human Behaviour, pending minor revisions to satisfy these requests and to comply with our editorial and formatting guidelines.

In particular, we ask that you address all remaining concerns by Reviewer #2 by:

- 1) contextualizing 'feature-based attention' along the lines of cognitive engagement/control;
- 2) rephrasing and toning down claims related to the spatial specificity of the findings; and
- 3) adding more information about the eye movements analysis.

We are now performing detailed checks on your paper and will send you a checklist detailing our editorial and formatting requirements within two weeks. Please do not upload the final materials and make any revisions until you receive this additional information from us.

Sincerely,

Giacomo Ariani
Editor
Nature Human Behaviour

Reviewer #1 (Remarks to the Author):

The authors carefully addressed all my questions and comments from the first round of reviews. I think the manuscript has improved considerably. It is well written and the presentation of the empirical data is very clear. I'm in favour of it being published in Nature Human Behaviour in its

current form.

Reviewer #2 (Remarks to the Author):

I thank the authors for their careful consideration of my comments. They have for example gathered substantial additional data for the control experiment, shown more of the whole-brain data, have run new analyses of the eye movement amplitude data, run a Bayes Factor test of the null result in the control experiment, and contacted the editor to discuss one of the points. However some of my major concerns have not been allayed sufficiently and there is still not enough support for the central claim that the current data indicates that the IFJ has a key role in attention.

My first main concern was that there was little basis for describing any effects here as “attentional”. The authors’ responses do not directly address this, but rather only indirectly. For example they argue that previous work has found the IFJ to be involved in attention; that the task was hard (75% difficult) making it unlikely that participants could track both stimuli at once; that effects in BOLD and EEG and DDM mirror what is seen in other studies attentional tasks. However this is not sufficient for invoking “attention” in the current dataset, which requires a direct behavioural implementation demonstrating that selection of some material over others is taking place in a way that enhances performance. Otherwise it is not clear that participants really did prioritise the processing as hoped. To address this concern this would require a fundamental dilution of the interpretation.

Secondly I argued that the spatial resolution of EEG and tACS are not sufficient to specifically implicate the IFJ as opposed to neighbouring brain regions. The authors’ response is to show that the beamforming modelling of the EEG results closely matches the fMRI and to interpret the e-field modelling analysis as suggesting a relatively focal IFJ effect. But the field modelling actually implicates a very large swathe of brain in the Suppl Fig 7 and the EEG data in Suppl Fig 5 that is supposed to support the fMRI data actually shows (in green) activations medial, lateral and anterior to the IFJ fMRI activation. This does not support that the effects can be selectively associated with the IFJ as opposed to other brain areas.

Thirdly, for a key new eye movement analysis, it is reported that “The density of amplitudes show that there are no major differences in amplitudes between the motion, scene, non attentive, in-phase or out-of-phase trials.” Please add more information about how this was calculated and what criteria is used to define “major”. For example in the new figure R2 part c, the second panel from the left shows the density of the saccade eye movement amplitudes and big differences seem apparent between the yellow, light blue and dark blue curves. In general it is unclear which datasets are plotted here in this panel, perhaps because they are superimposed e.g. are the right hand panels within this R2 c both brown because all datasets overlap?

Reviewer #3 (Remarks to the Author):

The authors addressed all comments in detail. I have no remaining queries.

Reviewer #1

Remarks to the Author:

The authors carefully addressed all my questions and comments from the first round of reviews. I think the manuscript has improved considerably. It is well written and the presentation of the empirical data is very clear. I'm in favour of it being published in Nature Human Behaviour in its current form.

Resp: We thank the reviewer for all their input and are happy they are in favour of publishing our paper.

Reviewer #2:

Remarks to the Author:

I thank the authors for their careful consideration of my comments. They have for example gathered substantial additional data for the control experiment, shown more of the whole-brain data, have run new analyses of the eye movement amplitude data, run a Bayes Factor test of the null result in the control experiment, and contacted the editor to discuss one of the points. However some of my major concerns have not been allayed sufficiently and there is still not enough support for the central claim that the current data indicates that the IFJ has a key role in attention.

1) *My first main concern was that there was little basis for describing any effects here as “attentional”. The authors’ responses do not directly address this, but rather only indirectly. For example they argue that previous work has found the IFJ to be involved in attention; that the task was hard (75% difficult) making it unlikely that participants could track both stimuli at once; that effects in BOLD and EEG and DDM mirror what is seen in other studies attentional tasks. However this is not sufficient for invoking “attention” in the current dataset, which requires a direct behavioural implementation demonstrating that selection of some material over others is taking place in a way that enhances performance. Otherwise it is not clear that participants really did prioritise the processing as hoped. To address this concern this would require a fundamental dilution of the interpretation.*

Resp: We have added a paragraph in the discussion to contextualize feature-based attention along with other control mechanisms (Page 8, 5th paragraph):

‘On a similar note, studies have shown the prefrontal cortex is involved in multiple control mechanisms with considerable overlap both in the involved brain structures as well as the mechanisms. Notable examples are attention and working memory [Zhou2022, Panichello2021]. One idea is that working memory is also a selection mechanism, selecting behaviorally important items to facilitate manipulation and recollection of the information. We find that our electrical manipulation of the IFJ has an effect on behavior through the attended visual evidence, but not the unattended visual evidence. Therefore, it remains as a possibility that our non-invasive stimulation intervention did not uniquely increase top-down attention per se, but the observed behavioral modulations might also be reflected in

modulations of short term working memory control. It will be interesting to investigate in future work possible dissociations in the mechanisms of prefrontal control affected by the non(invasive) neuromodulation.'

2) *Secondly I argued that the spatial resolution of EEG and tACS are not sufficient to specifically implicate the IFJ as opposed to neighbouring brain regions. The authors' response is to show that the beamforming modelling of the EEG results closely matches the fMRI and to interpret the e- field modelling analysis as suggesting a relatively focal IFJ effect. But the field modelling actually implicates a very large swathe of brain in the Suppl Fig 7 and the EEG data in Suppl Fig 5 that is supposed to support the fMRI data actually shows (in green) activations medial, lateral and anterior to the IFJ fMRI activation. This does not support that the effects can be selectively associated with the IFJ as opposed to other brain areas.*

Resp: We have toned down the claim about spatial specificity of the findings (Page 7, 3rd paragraph):

'... Lastly, although we have used the relatively focal ring electrode montage, and electric field modelling supports IFJ targeting (Supplementary Fig. 7), neighboring tissues around the IFJ are also influenced by the electric fields. We cannot rule out with certainty that we directly or indirectly stimulated other brain areas which could be causally involved in feature-based attention. Future advances in non-invasive neuromodulation selectivity could allow us to determine more precisely which areas are driving the results.'

3) *Thirdly, for a key new eye movement analysis, it is reported that "The density of amplitudes show that there are no major differences in amplitudes between the motion, scene, non attentive, in-phase or out-of-phase trials." Please add more information about how this was calculated and what criteria is used to define "major". For example in the new figure R2 part c, the second panel from the left shows the density of the saccade eye movement amplitudes and big differences seem apparent between the yellow, light blue and dark blue curves. In general it is unclear which datasets are plotted here in this panel, perhaps because they are superimposed e.g. are the right hand panels within this R2 c both brown because all datasets overlap?*

Resp: We have added information about the eye movements analysis (Supplementary Fig 6, page 5):

'The colors in this panel and the rest of the figure represent the dataset of which the gaze data was taken from: orange, blue, yellow, green and red indicate motion, scene, no- attention, in-phase and out-of-phase, respectively. ... The density of amplitudes show that the imaging results (fMRI and EEG) cannot be explained by differences in eye-movements. We found higher activity in the IFJ in the attentive compared to the non attentive condition, while amplitudes of eye movements are larger in the non-attentive condition (mean +/- SEM amplitude motion microsaccades = 0.42 +/- 0.01, image microsaccades = 0.42 +/- 0.01, no attention microsaccades = 0.41 +/- 0.01, motion saccades = 3.0 +/- 0.1, image saccades = 2.8 +/- 0.1, no attention saccades = 3.1 +/- 0.1). In the right two panels the distributions color is brown, since the red and green distribution completely overlap (mean +/- SEM amplitude in-phase microsaccades = 0.24 +/- 0.002, out-phase microsaccades = 0.24 +/- 0.002, in-phase saccades = 2.5 +/- 0.03, out-phase saccades = 2.5 +/- 0.03).'

Reviewer #3:

Remarks to the Author:

The authors addressed all comments in detail. I have no remaining queries.

Resp: Thanks to the reviewer for their thoughtful comments through the peer review process.

Final Decision Letter:

Dear Prof Polania,

We are pleased to inform you that your Article "Causal phase-dependent control of non-spatial attention in human prefrontal cortex", has now been accepted for publication in *Nature Human Behaviour*.

Please note that *Nature Human Behaviour* is a Transformative Journal (TJ). Authors may publish their research with us through the traditional subscription access route or make their paper immediately open access through payment of an article-processing charge (APC). Authors will not be required to make a final decision about access to their article until it has been accepted. Find out more about Transformative Journals

With best regards,

Giacomo Ariani
Editor
Nature Human Behaviour